# Uniparental nuclear inheritance following bisexual mating in fungi

Vikas Yadav, Sheng Sun, Joseph Heitman*

Department of Molecular Genetics and Microbiology, Duke University Medical Center, Durham, United States

**Abstract** Some remarkable animal species require an opposite-sex partner for their sexual development but discard the partner's genome before gamete formation, generating hemi-clonal progeny in a process called hybridogenesis. Here, we discovered a similar phenomenon, termed pseudosexual reproduction, in a basidiomycete human fungal pathogen, *Cryptococcus neoformans*, where exclusive uniparental inheritance of nuclear genetic material was observed during bisexual reproduction. Analysis of strains expressing fluorescent reporter proteins revealed instances where only one of the parental nuclei was present in the terminal sporulating basidium. Whole-genome sequencing revealed that the nuclear genome of the progeny was identical with one or the other parental genome. Pseudosexual reproduction was also detected in natural isolate crosses where it resulted in mainly *MAT*α progeny, a bias observed in *Cryptococcus* ecological distribution as well. The mitochondria in these progeny were inherited from the *MAT***a** parent, resulting in nuclear-mitochondrial genome exchange. The meiotic recombinase Dmc1 was found to be critical for pseudosexual reproduction. These findings reveal a novel, and potentially ecologically significant, mode of eukaryotic microbial reproduction that shares features with hybridogenesis in animals.

## Introduction

Most multicellular organisms in nature undergo (bi)sexual reproduction involving two partners of the opposite sex to produce progeny. In most cases, following the fusion of the two haploid gametes, the diploid zygote receives one copy of the genetic material from each parent. To produce these haploid gametes, a diploid germ cell of the organism undergoes meiosis, which involves recombination between the two parental genomes, generating recombinant product. Recombination confers benefits by bringing together beneficial mutations and segregating away deleterious ones (*Dimijian, 2005*; *Meirmans, 2009*). In contrast, some organisms undergo variant forms of sexual reproduction, including parthenogenesis, gynogenesis, androgenesis, and hybridogenesis, and in doing so, produce clonal or hemi-clonal progeny (*Avise, 2015*; *Neaves and Baumann, 2011*).

In parthenogenesis, a female produces clonal progeny from its eggs without any contribution from a male partner (*Avise, 2015*; *Horandl, 2009*). Gynogenesis and androgenesis occur when the fusion of an egg with a sperm induces cell division to produce clonal female or male zygotes, respectively (*Lehtonen et al., 2013*). During hybridogenesis, an egg from one species fuses with the sperm from another species to generate a hybrid diploid zygote (*Lavanchy and Schwander, 2019*). However, one of the parental genomes is excluded during development, in a process termed genome exclusion that occurs before gametogenesis. The remaining parental genome undergoes replication followed by meiosis to produce an egg or a sperm. The sperm or egg then fuses with an opposite-sex gamete to generate a hemiclonal progeny. Because only one parent contributes genetic material to the progeny, but both parents are physically required, this phenomenon has been termed sexual parasitism (*Lehtonen et al., 2013*; *Umphrey, 2006*). While most of the reported cases of hybridogenesis are from female populations, recent reports suggest that it may also occur in male populations of some species (*Doležálková et al., 2016*; *Schwander and Oldroyd, 2016*). Currently,

*For correspondence:
heitm001@duke.edu

Competing interests: The authors declare that no competing interests exist.

**eLife digest** Sexual reproduction enables organisms to recombine their genes to generate progeny that have higher levels of evolutionary fitness. This process requires reproductive cells – like the sperm and egg – to fuse together and mix their two genomes, resulting in offspring that are genetically distinct from their parents.

In a disease-causing fungus called *Cryptococcus neoformans*, sexual reproduction occurs when two compatible mating types (*MAT***a** and *MAT*α) merge together to form long branched filaments called hyphae. Cells in the hyphae contain two nuclei – one from each parent – which fuse in specialized cells at the end of the branches called basidia. The fused nucleus is then divided into four daughter nuclei, which generate spores that can develop into new organisms. In nature, the mating types of *C. neoformans* exhibit a peculiar distribution where *MAT*α represents 95% or more of the population. However, it is not clear how this fungus successfully reproduces with such an unusually skewed distribution of mating types.

To investigate this further, Yadav et al. tracked the reproductive cycle of *C. neoformans* applying genetic techniques, fluorescence microscopy, and whole-genome sequencing. This revealed that during hyphal branching some cells lose the nucleus of one of the two mating types. As a result, the nuclei of the generated spores only contain genetic information from one parent.

Yadav et al. named this process pseudosexual reproduction as it defies the central benefit of sex, which is to produce offspring with a new combination of genetic information. Further experiments showed that this unconventional mode of reproduction can be conducted by fungi isolated from both environmental samples and clinical patient samples. This suggests that pseudosexual reproduction is a widespread and conserved process that may provide significant evolutionary benefits.

*C. neoformans* represents a flexible and adaptable model organism to explore the impact and evolutionary advantages of sex. Further studies of the unique reproductive strategies employed by this fungus may improve the understanding of similar processes in other eukaryotes, including animals and plants. This research may also have important implications for understanding and controlling the growth of other disease-causing microbes.

hybridogenesis has only been observed in the animal kingdom in some species of frogs, fishes, and snakes. Plants also exhibit parthenogenesis (aka apomixis), along with gynogenesis and androgenesis (*Lehtonen et al., 2013*; *Mirzaghaderi and Hörandl, 2016*).

Unlike animals, most fungi do not have sex chromosomes; instead, cell-type identity is defined by the mating-type (*MAT*) locus (*Heitman, 2015*; *Heitman et al., 2013*). While many fungi are heterothallic, with opposite mating types in different individuals, and undergo sexual reproduction involving two partners of compatible mating types, other fungi are homothallic, with opposite mating types residing within the same organism, and can undergo sexual production during solo culture in the absence of a mating partner. One class of homothallic fungi undergoes unisexual reproduction, during which cells of a single mating type undergo sexual reproduction to produce clonal progeny, similar to parthenogenesis (*Heitman, 2015*; *Lee et al., 2010*). Gynogenesis and hybridogenesis have not been identified in the fungal kingdom thus far.

*Cryptococcus neoformans* is a basidiomycete human fungal pathogen that exists as either one of two mating types, *MAT***a** or *MAT*α (*Sun et al., 2019a*). During sexual reproduction, two haploid yeast cells of opposite mating types interact and undergo cell-cell fusion (*Kwon-Chung, 1975*; *Kwon-Chung, 1976*; *Sun et al., 2019b*). The resulting dikaryotic zygote then undergoes a morphological transition and develops into hyphae whose termini mature to form basidia. In the basidium, the two parental nuclei fuse (karyogamy), and the resulting diploid nucleus undergoes meiosis to produce four daughter nuclei (*Idnurm, 2010*; *Kwon-Chung, 1976*; *Sun et al., 2019b*; *Zhao et al., 2019*). These four haploid nuclei repeatedly divide via mitosis and bud from the surface of the basidium to produce four long spore chains. Interestingly, in addition to this canonical heterothallic sexual reproduction, a closely related species, *C. deneoformans* can undergo unisexual reproduction (*Lin et al., 2005*; *Roth et al., 2018*; *Sun et al., 2014*).

In a previous study, we generated a genome-shuffled strain of *C. neoformans*, VYD135α, by using the CRISPR-Cas9 system targeting centromeric transposons in the lab strain H99α. This led to multiple centromere-mediated chromosome arm exchanges in strain VYD135α when compared to the parental strain H99α, without any detectable changes in gene content between the two genomes (*Yadav et al., 2020*). In addition, strain VYD135α exhibits severe sporulation defects when mated with strain KN99**a** (which is congenic with strain H99α but has the opposite mating type), likely due to the extensive chromosomal rearrangements introduced into the VYD135α strain. In this study, we show that the genome-shuffled strain VYD135α can in fact produce spores in crosses with *MAT***a** *C. neoformans* strains after prolonged incubation. Analysis of these spores reveals that the products from each individual basidium contain genetic material derived from only one of the two parents. Whole-genome sequencing of the progeny revealed an absence of recombination between the two parental genomes. The mitochondria in these progeny were found to always be inherited from the *MAT***a** parent, consistent with known mitochondrial uniparental inheritance (UPI) patterns in *C. neoformans* (*Sun et al., 2020a*). Using strains with differentially fluorescently labeled nuclei, we discovered that in a few hyphal branches as well as in basidia, only one of the two parental nuclei was present and produced spores, leading to uniparental nuclear inheritance. We also observed the occurrence of such uniparental nuclear inheritance in wild-type and natural isolate crosses. Furthermore, we found that the meiotic recombinase Dmc1 plays a central role during this unusual mode of reproduction of *C. neoformans*. Overall, this mode of sexual reproduction of *C. neoformans* exhibits striking parallels with hybridogenesis in animals.

## Results

### Chromosomal translocation strain exhibits unusual sexual reproduction

Previously, we generated a strain (VYD135α) with eight centromere-mediated chromosome translocations compared to the wild-type parental isolate H99α (*Yadav et al., 2020*). Co-incubation of the wild-type strain KN99**a** with the genome-shuffled strain VYD135α resulted in hyphal development and basidia production, but no spores were observed during a standard 2-week incubation. However, when sporulation was assessed at later time points in the VYD135α×KN99**a** cross, we observed a limited number of sporulating basidia (16/1201=1.3%) after 5 weeks compared to a much greater level of sporulation in the wild-type H99α×KN99**a** cross (524/599=88%) (*Figure 1A–D*). None of these strains exhibited any filamentation on their own even after 5 weeks of incubation, indicating that the sporulation events were not a result of unisexual reproduction (*Figure 1A–B*). To analyze this delayed sporulation process in detail, spores from individual basidia were dissected and germinated to yield viable F1 progeny. As expected, genotyping of the mating-type locus in the H99α×KN99**a** progeny revealed that both *MAT***a** and *MAT*α progeny were produced from each basidium (*Figure 1E and G*, *Table 1*). In contrast, the same analysis for VYD135α×KN99**a** revealed that all germinating progeny from each individual basidium possessed either only the *MAT*α or the *MAT***a** allele (*Figure 1E and G*, *Table 1*). Polymerase chain reaction (PCR) assays also revealed that the mitochondria in all of these progeny were inherited from the *MAT***a** parent, in accord with known UPI (*Figure 1F–G*). These results suggest the inheritance of only one of the parental nuclei in the VYD135α×KN99**a** F1 progeny. The presence of mitochondria from only the *MAT***a** parent in *MAT*α progeny further confirmed that these progeny were the products of fusion between the parent strains and were not the products of unisexual reproduction.

### Fluorescence microscopy reveals uniparental nuclear inheritance after mating

Next, we tested whether the uniparental inheritance detected at the *MAT* locus also applied to the entire nuclear genome. To address this, we established a fluorescence-based assay in which the nuclei of strains H99α and VYD135α were labeled with GFP-H4, whereas the KN99**a** nucleus was marked with mCherry-H4. In a wild-type cross (H99α×KN99**a**), the nuclei in the hyphae as well as in the spores were yellow to orange because both nuclei were in a common cytoplasm and thus incorporated both the GFP-tagged and the mCherry-tagged histone H4 proteins (*Figure 2—figure supplement 1A and B*). We hypothesized that in the cases of uniparental nuclear inheritance, only one

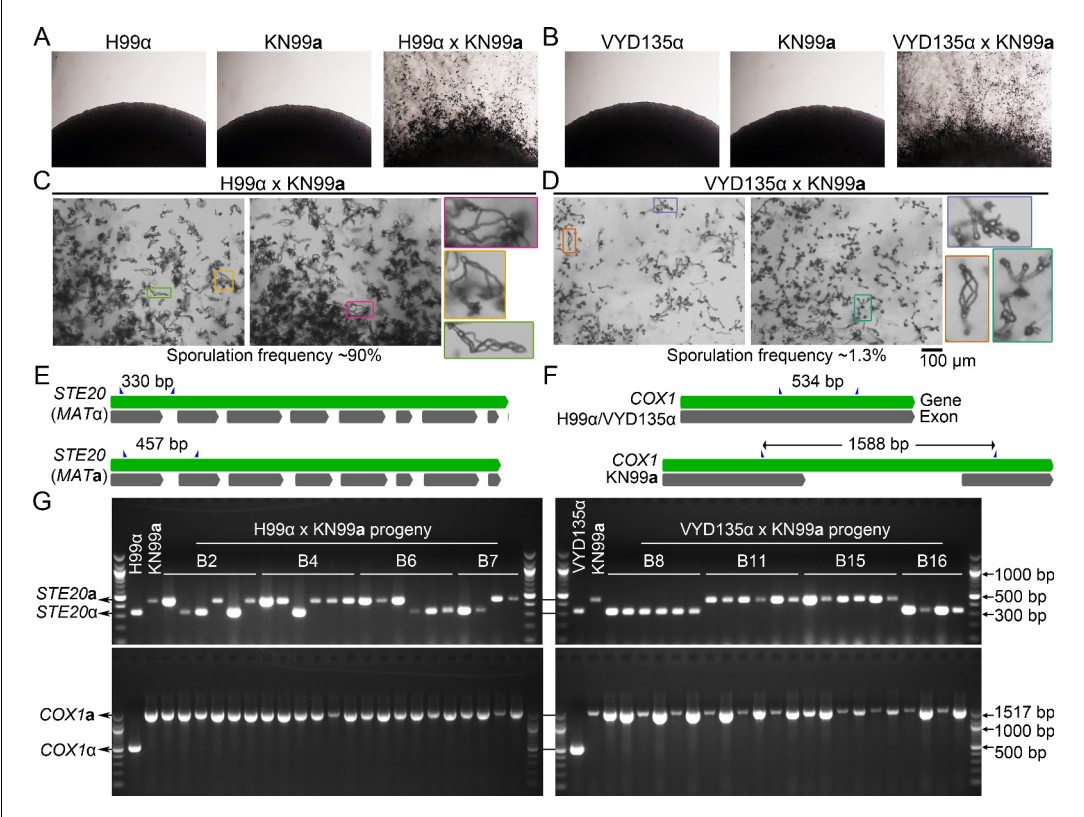

**Figure 1.** Chromosome shuffled strain exhibits unusual sexual reproduction. (**A, B**) Images of cultures for the individual strains H99α, KN99**a**, and VYD135α, showing no self-filamentation on mating medium. Magnification=10×. (**C, D**) Light microscopy images showing robust sporulation in the H99α×KN99**a** cross, whereas the VYD135α×KN99**a** cross exhibited robust hyphal development but infrequent sporulation events. The inset images in colored boxes show examples of basidia observed in each of the crosses. Scale bar, 100 μm. (**E, F**) A scheme showing the *MAT*α (H99α and VYD135α) and *MAT***a** (KN99**a**) alleles at the *STE20* (**E**) and *COX1* (**F**) loci. Primers used for PCR analysis are marked by blue triangles. (**G**) Gel images showing PCR amplification of *STE20* and *COX1* alleles in the progeny obtained from four different basidia for both H99α×KN99**a** and VYD135α×KN99**a** crosses. PCR analysis for the parental strains is also shown, and key bands for DNA marker are labeled. PCR, polymerase chain reaction.

of the nuclei would reach the terminal basidium and would thus harbor only one fluorescent nuclear color signal (*Figure 2—figure supplement 1A*).

After establishing this fluorescent tagging system using the wild-type strains H99α×KN99**a**, shuffled-strain VYD135α×KN99**a** crosses with fluorescently labeled strains were examined. In the wild-type cross, most of the basidia formed robust spore chains with both fluorescent colors observed in them, while a small population (~1%) of basidia exhibited spore chains with only one color, representing uniparental nuclear inheritance (*Figure 2A* and *Figure 2—figure supplement 2A*). In contrast, the majority of the basidium population in the shuffled-strain VYD135α×KN99**a** cross did not exhibit sporulation, and the two parental nuclei appeared fused but undivided (*Figure 2B* and *Figure 2—figure supplement 2B*). A few basidia (~1%) bore spore chains with only one fluorescent color, marking uniparental nuclear inheritance events. While the basidia with uniparental nuclear inheritance in the H99α×KN99**a** cross were a small fraction (~1%) of sporulating basidia, the uniparental basidia accounted for all of the sporulating basidia in the VYD135α×KN99**a** cross. Taken together, these results show that the uniparental nuclear inheritance leads to the generation of clonal progeny but requires mating, the cell-cell fusion between parents of two opposite mating types. Thus, this process defies the main purpose of sexual reproduction, which is to produce recombinant progeny from two parents. Based on these observations, we define the process of uniparental nuclear inheritance during sporulation in *C. neoformans* as pseudosexual reproduction (and it is referred to as such hereafter). The progeny obtained via this process will be referred to as the uniparental progeny because they inherit a nuclear genome derived from only one of the two parents.

**Table 1.** Genotype analysis of basidia-specific spores germinated from H99α×KN99**a** and VYD135α×KN99**a** crosses.

| | H99α×KN99**a** cross | | | | VYD135α×KN99**a** cross | | | |
|---|---|---|---|---|---|---|---|---|
| Basidia # | Spores germinated/ dissected | % Germinated | *MAT* | Mito | Spores germinated/ dissected | % Germinated | *MAT* | Mito |
| 1 | 5/14 | 36 | 4α+1a | a | 12/24 | 50 | All α | a |
| 2 | 14/14 | 100 | 7α+7a | a | 6/10 | 60 | All α | a |
| 3 | 12/14 | 86 | 2α+7a+3a/α | a | 15/15 | 100 | All a | a |
| 4 | 10/14 | 71 | 4α+6a | a | 22/27 | 81 | All a | a |
| 5 | 7/13 | 54 | 6a+1a/α | a | 3/12 | 25 | All α | a |
| 6 | 13/14 | 93 | 6α+7a | a | 25/27 | 93 | All α | a |
| 7 | 11/14 | 79 | 6α+5a | a | 4/4 | 100 | All α | a |
| 8 | 14/14 | 100 | 12α+2a | a | 10/13 | 77 | All α | a |
| 9 | 10/14 | 71 | 4α+6a | a | 13/15 | 87 | All α | a |
| 10 | 14/14 | 100 | 7α+7a | a | 31/61 | 51 | All α | a |
| 11 | 14/14 | 100 | 10α+4a | a | 10/10 | 100 | All a | a |
| 12 | 12/14 | 86 | 8α+4a | a | 4/5 | 80 | All a | a |
| 13 | 4/11 | 36 | All a | a | 24/28 | 86 | All a | a |
| 14 | 13/13 | 100 | 8α+5a | a | 16/28 | 57 | All a | a |
| 15 | 14/14 | 100 | 7α+7a | a | 11/11 | 100 | All a | a |
| 16 | 14/14 | 100 | 6α+8a | a | 10/22 | 45 | All α | a |

Mito refers to Mitochondria.

## Pseudosexual reproduction also occurs in natural isolates

After establishing the pseudosexual reproduction of lab strains, we sought to determine whether such events also occur with natural isolates. For this purpose, we selected two wild-type natural isolates, Bt63**a** and IUM96-2828**a** (referred to as IUM96**a** hereafter) (*Desjardins et al., 2017*; *Keller et al., 2003*; *Litvintseva et al., 2003*). IUM96**a** belongs to the same lineage as H99α/KN99**a** (VNI) and exhibits approximately 0.1% genome divergence from the H99α reference genome. Bt63**a** belongs to a different lineage of the *C. neoformans* species (VNBI) and exhibits ~0.5% genetic divergence from the H99α/KN99**a** genome. Both the Bt63**a** and the IUM96**a** genomes exhibit one reciprocal chromosome translocation with H99α, and as a result, share a total of 10 chromosome-level changes with the genome-shuffled strain VYD135α (*Figure 3A*). None of these strains are self-filamentous even after prolonged incubation on mating media but both cross efficiently with H99α and VYD135α (*Figure 3—figure supplement 1A*).

The H99α×Bt63**a** strains crossed rapidly (within a week) producing robust sporulation from most of the basidia observed. The VYD135α×Bt63**a** cross underwent a low frequency of sporulation (12 spore-producing basidia/840 basidia=1.4%) in 2–3 weeks (*Figure 3—figure supplement 1B*). Dissection of spores from the H99α×Bt63**a** cross revealed a low germination frequency (average of 25%) with two of the basidia showing no spore germination at all (*Supplementary file 1a*). This result is consistent with previous results, and the low germination frequency could be explained by the genetic divergence between the two strains (*Morrow et al., 2012*). Genotyping of germinated spores from the H99α×Bt63**a** cross revealed both *MAT***a** and *MAT*α progeny from individual basidia, with almost 75% of the meiotic events generating progeny that were heterozygous for the *MAT* locus (*Figure 3—figure supplement 1C* and *Supplementary file 1a*). For the VYD135α×Bt63**a** cross, spores from 15/20 basidia germinated and displayed a higher germination frequency than the H99α×Bt63**a** cross (*Supplementary file 1a*). Interestingly, all germinated progeny harbored only the *MAT*α mating type, whereas the mitochondria were in all cases inherited from the *MAT***a** parent (*Figure 3—figure supplement 1C*). These results suggest that pseudosexual reproduction also occurs with Bt63**a** and accounts for the high germination frequency of progeny from the VYD135α×Bt63**a** cross. The occurrence of pseudosexual reproduction was also identified using the fluorescence-based assay with crosses between the GFP-H4 tagged VDY135α and mCherry-H4 tagged Bt63**a** strains (*Figure 3—figure supplement 2*).

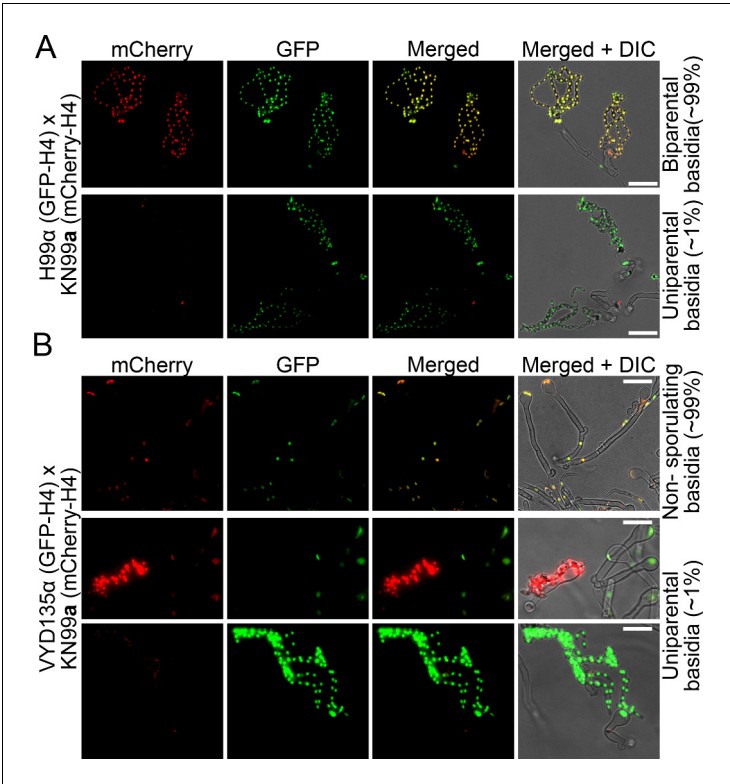

**Figure 2.** Fluorescence microscopy reveals uniparental nuclear inheritance in the wild-type crosses. (**A**) Crosses of GFP-H4 tagged H99α and mCherry-H4 tagged KN99**a** revealed the presence of both fluorescent markers in most spore chains along with uniparental nuclear inheritance in rare cases (~1%). In these few sporulating basidia, only one of the fluorescent signals was observed in the spore chains, reflecting the presence of only one parental nucleus in these basidia. (**B**) Crosses involving GFP-H4 tagged VYD135α and mCherry-H4 tagged KN99**a** revealed the presence of spore chains with only one fluorescent color. In the majority of basidia that have both parental nuclei, marked by both GFP and mCherry signals, spore chains are not produced, consistent with a failure of meiosis in these basidia. Scale bar, 10 μm.

The online version of this article includes the following figure supplement(s) for figure 2:

**Figure supplement 1.** Dynamics of sexual reproduction and sporulation analyzed with *C. neoformans* strains expressing nuclear-localized fluorescent reporter proteins.

**Figure supplement 2.** Nuclear dynamics during sporulation in the wild-type and VYD135α crosses.

Crosses with strain IUM96**a** also revealed a low level of sporulation (19/842=2.3%) with VYD135α but a high sporulation frequency with H99α (91%) (*Figure 3—figure supplement 1D*). Analysis of progeny from crosses involving IUM96**a** revealed a similar pattern to what was observed with crosses involving KN99**a**. The progeny from H99α×IUM96**a** exhibited variable basidium-specific germination frequencies and inherited both *MAT***a** and *MAT*α in each basidium, whereas VYD135α×IUM96**a** progeny from each basidium inherited exclusively either *MAT***a** or *MAT*α (*Figure 3—figure supplement 1E*, and *Supplementary file 1b*). Interestingly, we observed co-incident uniparental *MAT* inheritance and a high germination frequency in progeny of basidia 7, 8, and 9 from the H99α×IUM96**a** cross as well (*Figure 3—figure supplement 1E*, and *Supplementary file 1b*). Taken together, these results suggest that this unusual mode of sexual reproduction occurs with multiple natural isolates. We further propose that pseudosexual reproduction occurs in nature in parallel with canonical sexual reproduction.

## Uniparental progeny completely lack signs of nuclear recombination between the two parents

As mentioned previously, H99α (as well as the H99α-derived strain VYD135α) and Bt63**a** have approximately 0.5% genetic divergence. The occurrence of pseudosexual reproduction in the

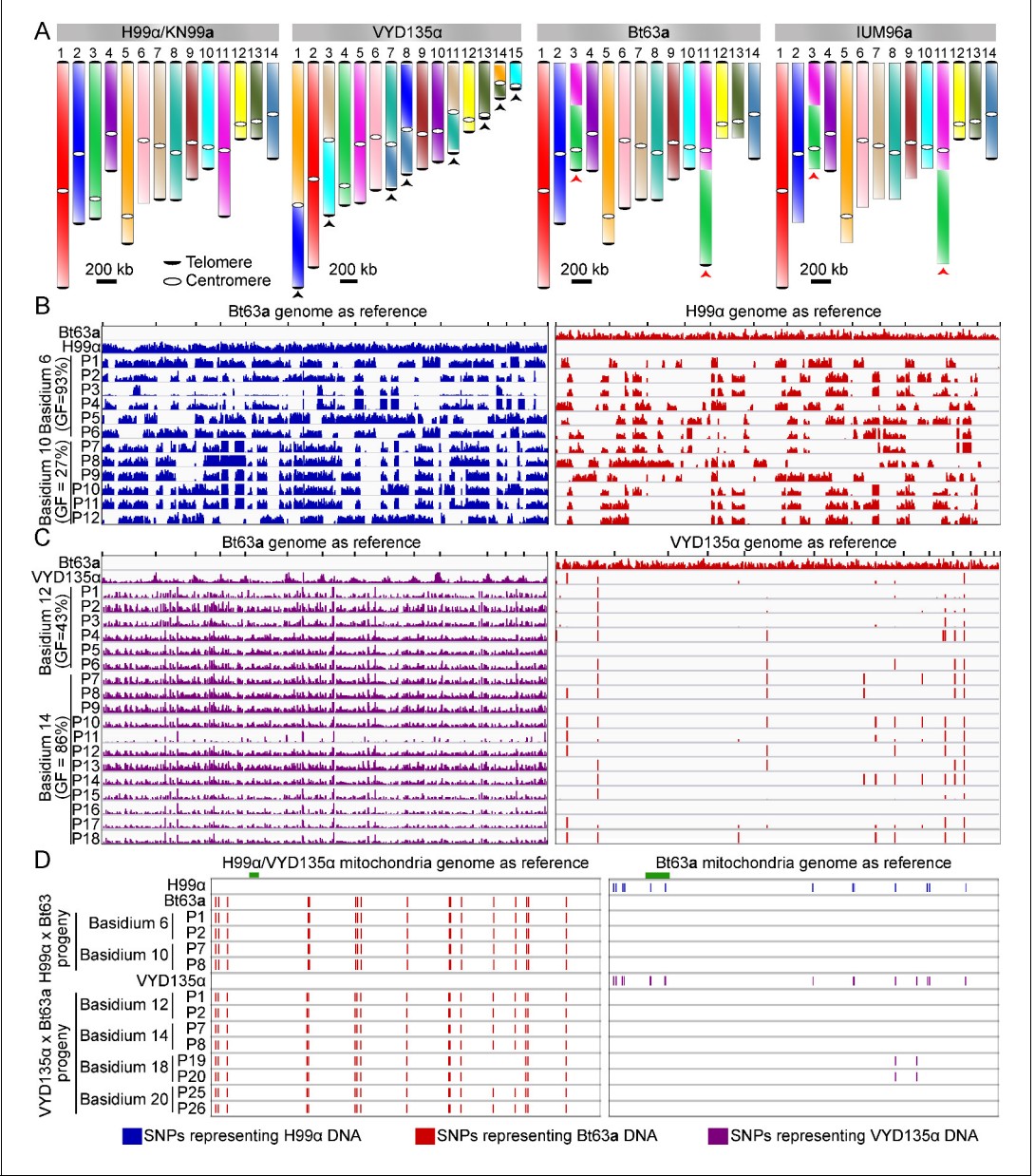

**Figure 3.** VYD135α progeny exhibit strict uniparental nuclear inheritance and lack the signature of meiotic recombination. (A) Chromosome maps for H99α/KN99**a**, VYD135α, Bt63**a**, and IUM96**a** showing the karyotype variation. The genome of the wild-type strain H99α served as the reference. Black arrowheads represent chromosome translocations between VYD135α and H99α whereas red arrowheads mark chromosomes with a translocation between H99α and Bt63**a** or IUM96**a**. (B) Whole-genome sequencing, followed by SNP identification, of H99α×Bt63**a** progeny revealed evidence of meiotic recombination in all of the progeny. The left panel shows SNPs with respect to the Bt63**a** genome whereas the right panel depicts SNPs against the H99α genome. H99α and Bt63**a** Illumina sequencing data served as controls for SNP calling. (C) SNP analysis of VYD135α ×Bt63**a** progeny revealed no contribution of the Bt63**a** parental genome in the progeny as evidenced by the presence of SNPs only against Bt63**a** (left panel) but not against the VYD135α genome (right panel). The presence of a few SNPs observed in VYD135α, as well as all VYD135α×Bt63**a** progeny, are within nucleotide repeat regions. GF stands for germination frequency and P stands for progeny. (D) SNP analysis of H99α×Bt63**a** and VYD135α×Bt63**a** progeny using mitochondrial DNA as the reference revealed that mitochondrial DNA is inherited from Bt63**a** in all of the progeny. Progeny obtained from VYD135α×Bt63**a** basidium 18 also revealed recombination between the two parental mitochondrial genomes as marked by the absence or presence of two SNPs when mapped against VYD135α and Bt63**a** mitochondrial genomes, respectively. The green bar in each panel depicts the locus used for PCR analysis of the mitochondrial genotype in the progeny. PCR, polymerase chain reaction; SNP, single nucleotide polymorphism.

The online version of this article includes the following figure supplement(s) for figure 3:

**Figure supplement 1.** Pseudosexual reproduction occurs in natural isolates, Bt63**a** and IUM96**a**.

**Figure supplement 2.** Bt63**a** fluorescence microscopy revealed pseudosexual reproduction events.

*Figure 3 continued on next page*

*Figure 3 continued*

**Figure supplement 3.** VYD135α×Bt63**a** progeny lack signatures of meiotic recombination.
**Figure supplement 4.** Mitochondria are inherited from *MAT***a** parent in all of the progeny.
**Figure supplement 5.** VYD135α×Bt63**a** progeny are haploid.
**Figure supplement 6.** IUM96**a** exhibits meiotic recombination in progeny with H99α but not with the genome shuffle strain VYD135α.
**Figure supplement 7.** Ploidy analysis of IUM96**a** progeny reveals haploid uniparental progeny.

VYD135α×Bt63**a** cross allowed us to test if the two parental genomes recombine with each other during development. We subjected progeny from crosses VYD135α×Bt63**a** and H99α×Bt63**a** to whole-genome sequencing. As expected, for the H99α×Bt63**a** cross, both parents contributed to the nuclear composition of their progeny, and there was clear evidence of meiotic recombination as determined by variant analysis (*Figure 3B*). However, when the VYD135α×Bt63**a** progeny were similarly analyzed, the nuclear genome in each of the progeny was found to be inherited exclusively from only the VYD135α parent (*Figure 3C* and *Figure 3—figure supplement 3*), and the progeny exhibited sequence differences across the entire Bt63**a** genome. In contrast, the mitochondrial genome was inherited exclusively from the Bt63**a** parent (*Figure 3D* and *Figure 3—figure supplement 4*), in accord with the PCR assay results discussed above. In addition, the whole-genome sequencing data also revealed that while most of the H99α×Bt63**a** progeny exhibited aneuploidy, the genome-shuffled strain VYD135α×Bt63**a** progeny were euploid (*Figure 3—figure supplement 5A and B*), and based on flow cytometry analysis, these uniparental progeny were haploid (*Figure 3—figure supplement 5C*).

The progeny from crosses involving IUM96**a** as the *MAT***a** partner were also sequenced. Similar to the Bt63**a** analysis, the H99α×IUM96**a** progeny exhibited signs of meiotic recombination, whereas the VYD135α ×IUM96**a** progeny did not (*Figure 3—figure supplement 6*). Congruent with the mating-type analysis, the progeny in each of the basidia exclusively inherited nuclear genetic material from only one of the two parents. Furthermore, the H99α×IUM96**a** progeny were found to be aneuploid for some chromosomes, while the VYD135α×IUM96**a** progeny were completely euploid (*Figure 3—figure supplement 7*). We also sequenced four progeny from basidium 7 from the H99α×IUM96**a** cross, which were suspected to be uniparental progeny based on mating-type PCRs. This analysis showed that all four progeny harbored only H99α nuclear DNA and had no contribution from the IUM96**a** nuclear genome, further supporting the conclusion that pseudosexual reproduction occurs in wild-type crosses (*Figure 3—figure supplement 6A*). Similar to other progeny, the mitochondria in these progeny were inherited from the *MAT***a** parent (*Figure 3—figure supplement 1E*, and *Supplementary file 1b*). Combined, these results affirm the occurrence of a novel mode of sexual reproduction in *C. neoformans,* which is initiated by the fusion of two strains of opposite mating types, but whose progeny inherit DNA exclusively from one parent.

## Pseudosexual reproduction stems from nuclear loss via hyphal branches

Fluorescence microscopy revealed that only one of the two parental nuclei undergoes meiosis and produces spores in approximately 1% of the total basidia population. Based on this finding, we hypothesized that the basidia with only one parental nucleus might arise due to nuclear segregation events during hyphal branching. To gain further insight into this process, the nuclear distribution pattern along the sporulating hyphae was studied. As expected, imaging of long hyphae in the wild-type cross revealed the presence of pairs of nuclei with both fluorescent markers along the length of the majority of hyphae (*Figure 4A*). In contrast, tracking of hyphae from basidia with spore chains in the genome-shuffled strain VYD135α×KN99**a** cross revealed hyphal branches with only one parental nucleus, which were preceded by a hyphum with both parental nuclei (*Figure 4B*, *Figure 4—figure supplement 1A and B*). Unfortunately, a majority of the hyphae (>30 independent hyphae) we tracked were embedded into the agar, and most of these could not be tracked to the point of branching. For some others, we were able to image the hyphal branching point where two nuclei separate from each other but were then either broken or did not have mature basidia on them (*Figure 4—figure supplement 1B*). In total, we observed seven events of nuclear loss at hyphal branching in independent experiments and were able to track two of them to observe sporulation or

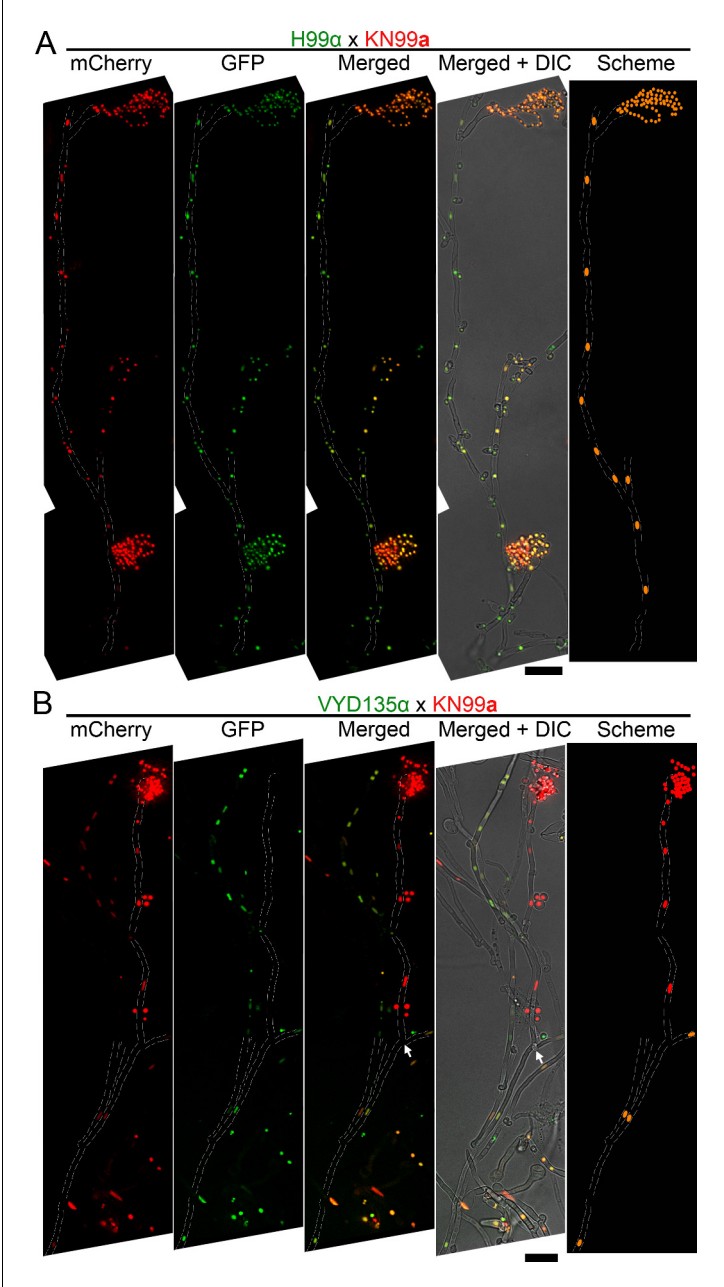

**Figure 4.** Pan-hyphal microscopy reveals the loss of one parental nucleus during pseudosexual reproduction. Spore-producing long hyphae were visualized in both (**A**) wild-type H99α×KN99**a** and (**B**) VYD135α×KN99**a** crosses to study the dynamics of nuclei in hyphae. Both nuclei were present across the hyphal length in the wild-type and resulted in the production of recombinant spores. On the other hand, one of the nuclei was lost during hyphal branching in the VYD135α×KN99**a** cross and resulted in uniparental nuclear inheritance in the spores that were produced. The arrow in (**B**) marks the hyphal branching point after which only one of the parental nuclei is present (also see *Figure 4—figure supplement 1A*). The images were captured as independent sections and assembled to obtain the final presented image. Scale bar, 10 μm.

The online version of this article includes the following figure supplement(s) for figure 4:

**Figure supplement 1.** Hyphal branches act as a gateway for nuclear separation during pseudosexual reproduction.

basidia formation at the tip. We also observed long hyphae with only one parental nucleus in the VYD135α×Bt63**a** cross as well, suggesting the mechanism might be similar between strains.

These results suggest that hyphal branching may facilitate the separation of one parental nucleus from the main hyphae harboring both parental nuclei. While this is the most plausible explanation based on our results, we cannot rule out other possible mechanisms, such as a role for clamp cells, leading to nuclear separation during hyphal growth. As a result, one of the parental genomes is excluded at a step before diploidization and meiosis, similar to the process of genome exclusion observed in hybridogenesis. We hypothesize that nuclear segregation can be followed by endoreplication occurring in these hyphal branches or in the basidium to produce a diploid nucleus that then ultimately undergoes meiosis and produces uniparental progeny, which will be explored in future studies.

## Meiotic recombinase Dmc1 is important for pseudosexual reproduction

Because the genomes of the uniparental progeny did not show evidence of meiotic recombination between the two parents, we tested whether pseudosexual reproduction involves meiosis. In addition, we sought to test our hypothesis that pseudosexual reproduction involves endoreplication that is followed by meiosis. We therefore tested whether Dmc1, a key component of the meiotic machinery, is required for pseudosexual reproduction. The meiotic recombinase gene *DMC1* was deleted in congenic strains H99α, VYD135α, and KN99**a**, and the resulting mutants were subjected to crossing. A previous report documented that *dmc1Δ* bilateral crosses (both the parents are mutant for *DMC1*) display significantly reduced, but not completely abolished, sporulation in *Cryptococcus* (*Lin et al., 2005*). We observed a similar phenotype with the H99α *dmc1Δ*×KN99**a** *dmc1Δ* cross. While most of the basidia were devoid of spore chains, a small percentage (21/760=2.7%) of the population bypassed the requirement for Dmc1 and produced spores (*Figure 5A* and *Figure 5— figure supplement 1A*). When dissected, the germination frequency for these spores was found to be very low (~22% on average) with spores from many basidia not germinating at all (*Supplementary file 1c*). Furthermore, *MAT*-specific PCRs revealed that some of the progeny were aneuploid or diploid. For VYD135α *dmc1Δ*×KN99**a** *dmc1Δ*, many fewer basidia (~0.1%) produced spore chains as compared to ~1% sporulation in VYD135α×KN99**a** (*Figure 5A,B* and *Figure 5—figure supplement 1B*). *dmc1* mutant unilateral crosses (one of the two parents is mutant and the other one is wild-type) sporulated at a frequency of 0.4% suggesting that only one of the parental strains was producing spores (*Figure 5B*). When a few sporulating basidia from the VYD135α *dmc1Δ*×KN99**a** *dmc1Δ* bilateral cross were dissected, two different populations of basidia emerged, one with no spore germination, and the other with a high spore germination frequency and uniparental *MAT* inheritance (*Supplementary file 1c*). We hypothesized that the basidia with a high spore

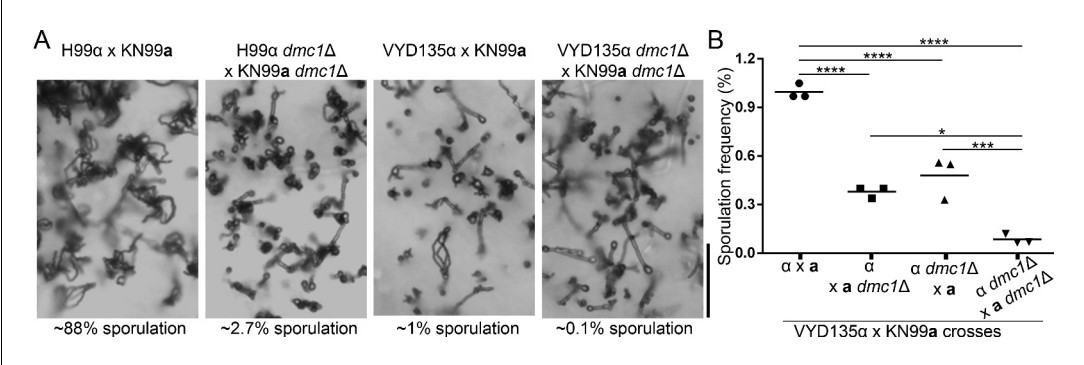

**Figure 5.** Meiotic recombinase Dmc1 is required for pseudosexual reproduction. (**A**) Light microscopy images showing the impact of *dmc1* mutation on sexual and pseudosexual reproduction in *C. neoformans*. Scale bar, 100 µm. (**B**) A graph showing quantification (n=3) of sporulation events in multiple crosses with *dmc1Δ* mutants. At least 3000 basidia were counted in each experiment.

The online version of this article includes the following figure supplement(s) for figure 5:

**Figure supplement 1.** Dmc1 deletion leads to severe sporulation defects in both sexual and pseudosexual reproduction.

**Figure supplement 2.** Meiotic regulator Dmc1 is required for pseudosexual reproduction.

germination frequency represent those that have escaped the normal requirement for Dmc1. Overall, the *DMC1* deletion led to a 20-fold reduction in viable sporulation in the VYD135α×KN99**a** cross, observed as a ten fold decrease from the number of sporulation events in the bilateral cross and a further two fold reduction in the number of basidia producing viable spores.

To further support these findings, *DMC1* was deleted in mCherry-H4 tagged KN99**a** and crossed with GFP-H4 tagged VYD135α. We hypothesized that GFP-H4 tagged VYD135α would produce spore chains in this cross because it harbors *DMC1*, whereas mCherry-H4 tagged KN99**a** *dmc1Δ* would fail to do so. Indeed, all 11 observed basidia with only the GFP-H4 fluorescence signal were found to produce spores, but only 2 out of 19 mCherry-H4 containing basidia exhibited sporulation (*Figure 5—figure supplement 2*). These results combined with the spore dissection findings show that Dmc1 is critical for pseudosexual reproduction. While these results provide concrete evidence for meiosis as a part of pseudosexual reproduction, they also suggest the occurrence of a preceding endoreplication event. However, further studies will need to be conducted to validate and confirm endoreplication or alternate mechanisms to achieve the ploidy necessary for a classical meiosis event.

## Discussion

Hybridogenesis and parthenogenesis are mechanisms that allow some organisms to overcome some hurdles of sexual reproduction and produce hemiclonal or clonal progeny (*Avise, 2015*; *Horandl, 2009*; *Lavanchy and Schwander, 2019*). However, harmful mutations are not filtered in these processes, making them disadvantageous during evolution and thus restricting the occurrence of these processes to a limited number of animal species (*Lavanchy and Schwander, 2019*). In this study, we discovered and characterized the occurrence of a phenomenon in fungi that resembles hybridogenesis and termed it pseudosexual reproduction (*Figure 6—figure supplement 1*). Fungi are known to exhibit asexual, (bi)sexual, unisexual, and parasexual reproduction, and can switch between these reproductive modes depending on environmental conditions (*Heitman, 2015*; *Heitman et al., 2013*). The discovery of pseudosexual reproduction further diversifies known reproductive modes in fungi, suggesting the presence of sexual parasitism in this kingdom.

Hybridogenesis in animals occurs between two different species. The result of hybridogenesis is the production of gametes that are clones of one of the parents, which then fuse with an opposite-sex gamete of the second species, generating hemiclonal offspring. In our study, we observed a similar phenomenon where only one parent contributes to spores, the counterpart of mammalian gametes. However, we observed this phenomenon occurring between different strains of the same species, *C. neoformans*. It is important to note that these strains vary significantly from each other in terms of genetic divergence and in one case by chromosome rearrangements to the extent that they could be considered different species. This suggests that hybridogenesis in animals and pseudosexual reproduction in fungi are similar to each other. Hybridogenesis requires the exclusion of one of the parents, which is followed by endoreplication of the other parent's genome and meiosis. The whole-genome sequence of the progeny in our study revealed the complete absence of one parent's genome, suggesting manifestations of genome exclusion during hyphal growth. The mechanism by which the retained parental genome increases its ploidy before meiosis remains to be further investigated in *C. neoformans*. Endoreplication is known to occur in the sister species *C. deneoformans* during unisexual reproduction, and we think that this is the most likely route via which ploidy is increased during pseudosexual reproduction.

The mechanism and time of genome exclusion during hybridogenesis in animals are not entirely understood, except for a few insights from diploid fishes of the genus *Poeciliopsis* and water frogs, *Pelophylax esculentus*. Studies using *Poeciliopsis* fishes showed that haploid paternal genome exclusion takes place during the onset of meiosis via the formation of a unipolar spindle, and thus, only the haploid set of maternal chromosomes is retained (*Cimino, 1972a*; *Cimino, 1972b*). On the other hand, studies involving *P. esculentus* revealed that genome exclusion occurs during mitotic division, before meiosis, which is followed by endoreplication of the other parental genome (*Heppich et al., 1982*; *Tunner and Heppich-Tunner, 1991*; *Tunner and Heppich, 1981*). A recent study, however, proposed that genome exclusion in *P. esculentus* could also take place during early meiotic phases (*Doležálková et al., 2016*). Using fluorescence microscopy, we examined the steps of nuclear exclusion in *C. neoformans* and found that it occurs during mitotic hyphal growth and not during meiosis.

We also observed that genome exclusion could happen with either of the two parents in *C. neoformans*, similar to what has also been reported for water frogs. However, for most other species, genome exclusion was found to occur with the male genome only, leaving behind the female genome for meiosis (*Cimino, 1972a*; *Holsbeek and Jooris, 2010*; *Lavanchy and Schwander, 2019*; *Umphrey, 2006*; *Uzzell et al., 1976*; *Vinogradov et al., 1991*). Multiple studies have shown the formation of meiotic synaptonemal complexes during hybridogenesis, clearly establishing the presence of meiosis during this process (*Dedukh et al., 2019*; *Dedukh et al., 2020*; *Nabais et al., 2012*). Our results showed that the meiotic recombinase Dmc1 is required for pseudosexual reproduction, suggesting the presence of meiosis, whereas there is no direct evidence for the role of a meiotic recombinase in hybridogenetic animals. Taken together, these results indicate that the mechanism might be at least partially conserved across distantly related species. Future studies will shed more light on this, and if established, the amenability of *C. neoformans* to genetic manipulation will aid in deciphering some of the unanswered questions related to hybridogenesis in animals.

The occurrence of pseudosexual reproduction might also have significant implications for *C. neoformans* biology. Most (>95%) of *Cryptococcus* natural isolates belong to only one mating type, α (*Zhao et al., 2019*). While the reason behind this distribution is unknown, one explanation could be the presence of unisexual reproduction in the sister species *C. deneoformans* and *C. gattii* (*Fraser et al., 2005*; *Lin et al., 2005*; *Phadke et al., 2014*). The presence of pseudosexual reproduction in *C. neoformans* might help explain the mating-type distribution pattern for this species. In this report, one of the *MAT***a** natural isolates, Bt63**a**, did not contribute to pseudosexual reproduction and the other isolate, IUM96**a**, produced uniparental progeny in only one basidium, while the rest of the basidia produced *MAT*α progeny. We hypothesized that *MAT***a** isolates may be defective in this process due to either a variation in their genomes or some other as yet undefined sporulation factor. As a result, pseudosexual reproduction could lead to the generation of predominantly α progeny in nature, reducing the *MAT***a** population and thus favoring the expansion of the α mating-type population. However, it is still possible that the preferential inheritance of the nuclear genome from one of the two parents is decided by genetic elements located in regions other than *MAT*, and whether the uniparental nuclear inheritance is mating-type specific remains to be elucidated. Furthermore, the occurrence of pseudosexual reproduction in other pathogenic species such as *C. deneoformans* and non-pathogenic species such as *C. amylolentus* will be investigated in future studies. Attempts to identify the occurrence of pseudosexual reproduction between species where hybrids are known to occur, *C. neoformans* and *C. deneoformans* hybrids, will also be made. These studies will help establish the scope of pseudosexual reproduction in *Cryptococcus* species and could be extended to other basidiomycetes.

We propose that pseudosexual reproduction can occur between any two opposite mating-type strains as long as each of them is capable of undergoing cell-cell fusion and at least one of them can sporulate. We speculate that pseudosexual reproduction might play a key role in *C. neoformans* survival during unfavorable conditions. In conditions where two mating partners are fully compatible, pseudosexual reproduction will be mostly hidden and might not be important (*Figure 6*, top panel). However, when the two mating partners are partially incompatible or completely incompatible due to high genetic divergence or karyotypic variation, pseudosexual reproduction will be important (*Figure 6*, left, right, and bottom panels). For example, most of the basidia in H99α and Bt63**a** cross largely produce aneuploid and/or inviable progeny leading to unsuccessful sexual reproduction. However, a small yet significant proportion of the basidia generate clonal progeny that are viable and fit via pseudosexual reproduction. We hypothesized that these progeny will have a better chance of survival and find a suitable mating partner in the environment whereas, the unfit recombinant progeny might fail to do so. In nature, this might allow a new genotype/karyotype to not only survive but also expand and will prove advantageous. If a new genotype/karyotype had only the option of undergoing sexual reproduction, it might not survive, restricting the evolution of a new strain. Overall, this mode of pseudosexual reproduction might act as an escape path from genomic incompatibilities between two related isolates and allow them to produce spores for dispersal.

One of the key differences between pseudosexual reproduction and unisexual reproduction observed in the *Cryptococcus* species complex is the inheritance of mitochondrial DNA. While both unisexual and pseudosexual reproduction result in clonal progeny with respect to the nuclear genome, the mitochondria in pseudosexual reproduction are almost exclusively inherited from the *MAT***a** parent (*Figure 6—figure supplement 1*). This results in the exchange of mitochondrial DNA

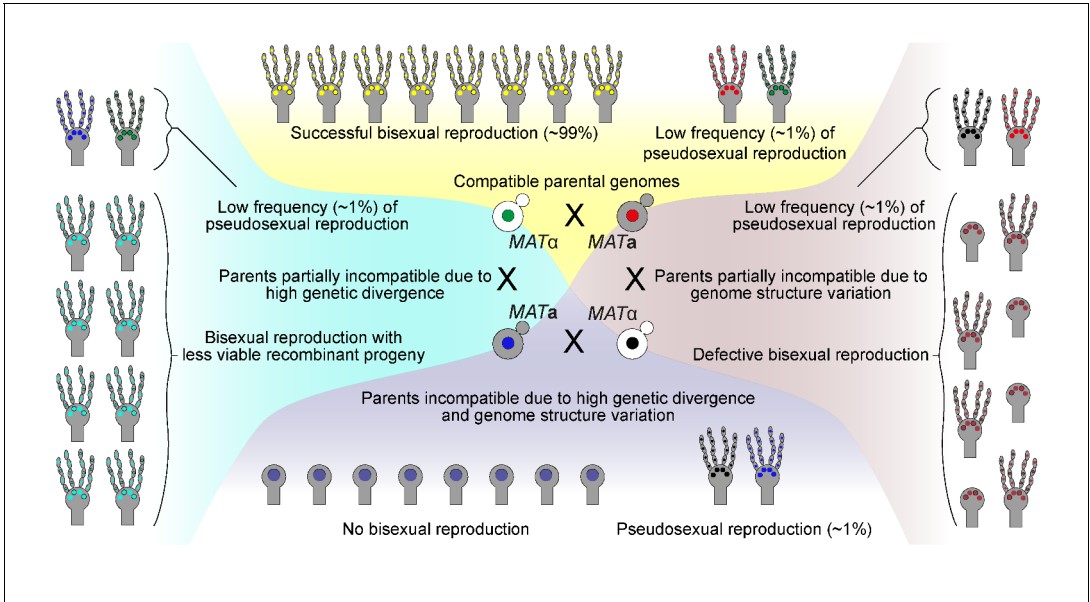

**Figure 6.** Model for the role of pseudosexual reproduction in *C. neoformans* ecology. Scenarios showing possible roles for pseudosexual reproduction under various hypothetical mating conditions. Except for one condition where the two parents are completely compatible with each other, pseudosexual reproduction could play a significant role in survival and dissemination despite its occurrence at a low frequency.

The online version of this article includes the following figure supplement(s) for figure 6:

**Figure supplement 1.** Unisexual, bisexual, and pseudosexual reproduction in *C. neoformans.*

in the progeny that inherit the *MATα* nuclear genome, resembling the nuclear-mitochondrial exchange observed during cytoduction in *Saccharomyces cerevisiae*. During cytoduction, mutants defective in nuclear fusion produce haploid progeny with nuclear genome from one parent, but a mixture of both parents cytoplasm resulting in the inheritance of one parental mitochondrial genome with the other parent's nuclear genome (*Conde and Fink, 1976*; *Lancashire and Mattoon, 1979*; *Zakharov and Yarovoy, 1977*). This process was used to study mitochondrial genetics with respect to the transfer of drug-resistance genes and other mitochondrial mutations. Similar to cytoduction, pseudosexual reproduction could be employed to study mitochondrial genetics, such as functional analysis of mitochondrial encoded drug resistance, and cytoplasmic inheritance of factors such as prions in *C. neoformans*.

The fungal kingdom is one of the more diverse kingdoms with approximately 3 million species (*Sun et al., 2020b*). The finding of hybridogenesis-like pseudosexual reproduction hints toward unexplored biology in this kingdom that might provide crucial clues for understanding the evolution of sex. Fungi have also been the basis of studies focused on understanding the evolution of meiosis, and the presence of genome reduction, as well as the parasexual cycle in fungi, have led to the proposal that meiosis evolved from mitosis (*Hurst and Nurse, 1991*; *Wilkins and Holliday, 2009*). Pseudosexual reproduction may be a part of an evolutionary process wherein genome exclusion followed by endoreplication and meiosis was an ancestral form of reproduction that preceded the evolution of sexual reproduction. Evidence supporting such a hypothesis can be observed in organisms undergoing facultative sex or facultative parthenogenesis (*Booth et al., 2012*; *Fields et al., 2015*; *Hodač et al., 2019*; *Hojsgaard and Horandl, 2015*). The presence of these organisms also suggests that a combination of both sexual and clonal modes of reproduction might prove to be evolutionarily advantageous.

## Materials and methods

### Strains and media

*C. neoformans* wild-type strains H99α and KN99**a** served as the wild-type isogenic parental lineages for the experiments (*Nielsen et al., 2003*; *Perfect et al., 1993*), in addition to *MAT***a** strains Bt63**a** and IUM96-2828**a** (*Keller et al., 2003*; *Litvintseva et al., 2003*). Strains were grown in YPD media for all experiments at 30°C unless stated otherwise. G418 and/or NAT were added at a final concentration of 200 and 100 μg/ml, respectively, for the selection of transformants. MS media was used for all the mating assays, which were performed as described previously (*Sun et al., 2019b*). Basidia-specific spore dissections were performed after 2–5 weeks of mating, and the spore germination frequency was scored after 5 days of dissection. All strains and primers used in this study are listed in *Supplementary file 1d* and *Supplementary file 1e*, respectively.

### Genotyping for mating-type locus and mitochondria

Mating type (*MAT*) and mitochondrial genotyping for all the progeny were conducted using PCR assays. Genomic DNA was prepared using the MasterPure Yeast DNA Purification Kit from Lucigen. To determine the *MAT*, the *STE20* allele present within the *MAT* locus was detected because it differs in length between the two different mating types. Primers specific to both *MAT***a** and *MAT*α (JOHE50979-50982 in *Supplementary file 1e*) were mixed in the same PCR mix, and the identification was made based on the length of the amplicon (*Figure 1E–G*). For the mitochondrial genotyping, the *COX1* allele present in the mitochondrial DNA was probed to distinguish between H99α/VYD135α and KN99**a**/IUM96**a**. For the differentiation between Bt63**a** and H99α/VYD135α, the *COB1* allele was used because *COX1* in Bt63**a** is identical to H99α/VYD135α. The difference for both *COX1* and *COB1* is the presence or absence of an intron and results in significantly different size products between *MAT*α and *MAT***a** parents (*Figure 1* and *Figure 3—figure supplement 1*). The primers used for these assays (JOHE51004-51007) are mentioned in *Supplementary file 1e*.

### Genomic DNA isolation for sequencing

Genomic DNA for whole-genome sequencing was prepared using the CTAB-based lysis method, as described previously (*Yadav et al., 2020*). Briefly, 50 ml of an overnight culture was pelleted, frozen at −80°C, and subjected to lyophilization. The lyophilized cell pellet was broken into a fine powder, mixed with lysis buffer, and the mix was incubated at 65°C for an hour with intermittent shaking. The mix was then cooled on ice, and the supernatant was transferred into a fresh tube, and an equal volume of chloroform (~15 ml) was added and mixed. The mix was centrifuged at 3200 rpm for 10 min, and the supernatant was transferred to a fresh tube. An equal volume of isopropanol (~18–20 ml) was added into the supernatant and mixed gently. This mix was incubated at −20°C for an hour and centrifuged at 3200 rpm for 10 min. The supernatant was discarded, and the DNA pellet was washed with 70% ethanol. The pellet was air-dried and dissolved in 1 ml of RNase containing 1× TE buffer and incubated at 37°C for 45 min. The DNA was again chloroform purified and precipitated using isopropanol, followed by ethanol washing, air drying, and finally dissolved in 200 μl 1× TE buffer. The DNA quality was estimated with NanoDrop, whereas DNA quantity was estimated with Qubit.

### Whole-genome Illumina sequencing, ploidy, and SNP analysis

Illumina sequencing of the strains was performed at the Duke sequencing facility core (https://genome.duke.edu/), using Novaseq 6000 as 150 paired-end sequencing. The Illumina reads, thus obtained, were mapped to the respective genome assembly (H99α, VYD135α, Bt63**a**, or IUM96**a**) using Geneious (RRID:SCR_010519) default mapper to estimate ploidy. The resulting BAM file was converted to a. tdf file, which was then visualized through IGV to estimate the ploidy based on read coverage for each chromosome.

For SNP calling and score for recombination in the progeny, Illumina sequencing data for each progeny was mapped to parental strain genome assemblies individually using the Geneious default mapper with three iterations. The mapped BAM files were used to perform variant calling using Geneious with 0.8 variant frequency parameter and at least 90× coverage for each variant. The variants thus called were exported as VCF files and imported into IGV for visualization purposes. H99α, Bt63**a**, IUM96**a**, and VYD135α Illumina reads were used as controls for SNP calling analysis.

## PacBio/Nanopore genome assembly and synteny comparison

To obtain high-molecular-weight DNA for Bt63**a** genome PacBio and IUM96**a** genome Nanopore sequencing, DNA was prepared as described above. The size estimation of DNA was carried out by electrophoresis of DNA samples using PFGE. For this purpose, the PFGE was carried out at 6 V/cm at a switching frequency of 1–6 s for 16 hr at 14°C. Samples with most of the DNA $\geq$100 kb or larger were selected for sequencing. For PacBio sequencing, the DNA sample was submitted to the Duke sequencing facility core. Nanopore sequencing was performed in our lab using a MinION device on an R9.4.1 flow cell. After sequencing, reads were assembled to obtain a Bt63**a** genome assembly via Canu (RRID:SCR_015880) using PacBio reads >2 kb followed by five rounds of pilon polishing (RRID: SCR_014731). For IUM96**a**, one round of nanopolish was also performed before pilon polishing. Once completed, the chromosomes were numbered based on their synteny with the H99α genome. For chromosomes involved in translocation (Chr 3 and Chr 11), the chromosome numbering was defined by the presence of the respective syntenic centromere from H99. Centromere locations were mapped based on BLASTn analysis with H99α centromere flanking genes.

Synteny comparisons between the genomes were performed with SyMAP v4.2 using default parameters (*Soderlund et al., 2011*) (http://www.agcol.arizona.edu/software/symap/). The comparison block maps were exported as .svg files and were then processed using Adobe Illustrator (RRID: SCR_010279) and Adobe Photoshop (RRID:SCR_014199) for representation purposes. The H99α genome was used as the reference for comparison purposes for plotting VYD135α, Bt63**a**, and IUM96**a** genomes. The centromere and telomere locations were manually added during the figure processing.

## Fluorescent tagging and microscopy

GFP and mCherry tagging of histone H4 were performed by integrating respective constructs at the safe haven locus (*Arras et al., 2015*). GFP-H4 tagging was done using the previously described construct, pVY3 (*Yadav and Sanyal, 2018*). For mCherry-H4 tagging, the GFP-containing fragment in pVY3 was excised using SacI and BamHI and was replaced with mCherry sequence PCR amplified from the plasmid pLKB25 (*Kozubowski and Heitman, 2010*). The constructs were then linearized using XmnI and transformed into desired strains using CRISPR transformation, as described previously (*Fan and Lin, 2018*). The transformants were screened by PCR, and correct integrants were obtained and verified using fluorescent microscopy.

To observe the fluorescence signals in the hyphae and basidia, a 2- to 3-week-old mating patch was cut out of the plate and directly inverted onto a coverslip in a glass-bottom dish. The dish was then used to observe filaments under a DeltaVision microscope available at the Duke University Light Microscopy Core Facility (https://microscopy.duke.edu/dv). The images were captured at 60× magnification with 2×2 bin size and z-sections of either 1 or 0.4 µm each. GFP and mCherry signals were captured using the GFP and mCherry filters in the Live-Cell filter set. The images were processed using Fiji-ImageJ (https://imagej.net/Fiji) (RRID:SCR_002285) and exported as tiff files as individual maximum projected images. The final figure was then assembled using Adobe Photoshop software for quality purposes.

## Sporulation frequency counting

To visualize hyphal growth and sporulation defects during mating assays, the mating plates were directly observed under a Nikon Eclipse E400 microscope. Hyphal growth and basidia images were captured using the top-mounted Nikon DXM1200F camera on the microscope. The images were processed using Fiji-ImageJ and assembled in Adobe Photoshop software.

For crosses involving wild-type H99α, VYD135α, KN99**a**, Bt63**a**, and IUM96**a**, approximately 1000 total basidia were counted after 4 weeks of mating, and the sporulation frequency was calculated. For crosses involving VYD135 *dmc1Δ* strain, three mating spots were setup independently. From each mating spot periphery, six images were captured after 3–4 weeks of mating. Basidia (both sporulating and non-sporulating) in each of these spots were counted manually after some processing of images using ImageJ. The sporulation frequency was determined by dividing the sporulating basidia by the total number of basidia for each spot. Each mating spot was considered as an independent experiment and at least 3000 basidia were counted from each mating spot.

## Flow cytometry

Flow cytometry analysis was performed as described previously (*Fu and Heitman, 2017*). Cells were grown on YPD medium for 2 days at 30°C, harvested, and washed with 1× phosphate-buffered saline buffer followed by fixation in 70% ethanol at 4°C overnight. Next, cells were washed once with 1 ml of NS buffer (10 mM Tris-HCl, pH=7.2, 250 mM sucrose, 1 mM EDTA, pH=8.0, 1 mM $MgCl_2$, 0.1 mM $CaCl_2$, 0.1 mM $ZnCl_2$, 0.4 mM phenylmethylsulfonyl fluoride, and 7 mM β-mercaptoethanol), and finally resuspended in 180 μl NS buffer containing 20 μl 10 mg/ml RNase and 5 μl 0.5 mg/ml propidium iodide (PI) at 37°C for 3–4 hr. Then, 50 μl stained cells were diluted in 2 ml of 50 mM Tris-HCl, pH=8.0, transferred to FACS compatible tube, and submitted for analysis at the Duke Cancer Institute Flow Cytometry Shared Resource. For each sample, 10,000 cells were analyzed on the FL1 channel on the Becton-Dickinson FACScan. Wild-type H99α and previously generated AI187 were used as haploid and diploid controls, respectively, in these experiments. Data analysis was performed using the FlowJo software (RRID:SCR_008520).

## Acknowledgements

The authors thank Shelby Priest and Arti Dumbrepatil for the critical reading of this manuscript. This study was supported by NIH/NIAID R01 award AI39115-24, R01 grant AI50113-16 awarded to JH, and R01 grant AI33654-04 awarded to JH, David Tobin, and Paul Magwene. JH is also Co-Director and Fellow of the CIFAR program *Fungal Kingdom: Threats and Opportunities*.

## Additional information

### Funding

| Funder | Grant reference number | Author |
|---|---|---|
| National Institute of Allergy and Infectious Diseases | AI50113-16 | Joseph Heitman |
| National Institute of Allergy and Infectious Diseases | AI39115-24 | Joseph Heitman |
| National Institute of Allergy and Infectious Diseases | AI33654-04 | Joseph Heitman |

The funders had no role in study design, data collection and interpretation, or the decision to submit the work for publication.

### Author contributions

Vikas Yadav, Conceptualization, Resources, Data curation, Formal analysis, Validation, Investigation, Visualization, Methodology, Writing - original draft, Writing - review and editing; Sheng Sun, Investigation, Methodology, Writing - review and editing; Joseph Heitman, Conceptualization, Supervision, Funding acquisition, Writing - original draft, Project administration, Writing - review and editing

### Author ORCIDs

Vikas Yadav  https://orcid.org/0000-0003-2650-9035
Sheng Sun  https://orcid.org/0000-0002-2895-1153
Joseph Heitman  https://orcid.org/0000-0001-6369-5995

### Decision letter and Author response

Decision letter https://doi.org/10.7554/eLife.66234.sa1
Author response https://doi.org/10.7554/eLife.66234.sa2

## Additional files

### Supplementary files

• Supplementary file 1. Genotyping of progeny obtained, strains and primers used for this study. (a). The genotype of basidia-specific spores dissected from H99α×Bt63**a** and VYD135α×Bt63**a** crosses. (b). The genotype of basidia-specific spores dissected from H99α×IUM96-2828**a** and VYD135α×IUM96-2828**a** crosses. (c). Genotype analysis of basidia-specific progeny from H99α *dmc1*Δ×KN99**a** *dmc1*Δ and VYD135α *dmc1*Δ×KN99**a** *dmc1*Δ crosses. (d). Strains used in this study. (e). Primers used in this study.

• Transparent reporting form

### Data availability

The sequence data generated in this study were submitted to NCBI with the BioProject accession number PRJNA682203.

The following dataset was generated:

| Author(s) | Year | Dataset title | Dataset URL | Database and Identifier |
|---|---|---|---|---|
| Yadav V, Sun S, Heitman J | 2020 | Uniparental reproduction in Cryptococcus neoformans | https://www.ncbi.nlm.nih.gov/bioproject/PRJNA682203 | NCBI BioProject, PRJNA682203 |

The following previously published datasets were used:

| Author(s) | Year | Dataset title | Dataset URL | Database and Identifier |
|---|---|---|---|---|
| Yadav V, Sun S, Coelho MA, Heitman J | 2020 | Illumina reads of VYD135 | https://www.ncbi.nlm.nih.gov/sra/?term=SRR10317030 | NCBI Sequence Read Archive, SRR10317030 |
| Broad Institute | 2012 | Illumina whole genome shotgun sequencing of genomic DNA paired-end library 'Pond-151755' containing sample 'Cryptococcus neoformans H99' | https://www.ncbi.nlm.nih.gov/sra/?term=SRR642222 | NCBI Sequence Read Archive, SRR642222 |
| Broad Institute | 2012 | Illumina whole genome shotgun sequencing of genomic DNA paired-end library 'Pond-151755' containing sample 'Cryptococcus neoformans H99' | https://www.ncbi.nlm.nih.gov/sra/?term=SRR647805 | NCBI Sequence Read Archive, SRR647805 |

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
