## [Decision Letter]

**Acceptance summary:**

Hybridogenesis is an unusual form of reproduction described in some animal species, that requires an opposite-sex partner although the partner's genome is discarded before gamete formation, producing hemi-clonal progeny. In this work, the authors characterize in the human fungal pathogen *Cryptococcus neoformans*, an unusual form of reproduction that exhibits striking parallels with hybridogenesis, and that is termed pseudosexual reproduction. The discovery of pseudosexual reproduction not only further expands the described reproductive modes in fungi, but it could help addressing yet unresolved mechanistic aspects of hybridogenesis in animals. This work will be of interest to the fungal community, and also to a broader audience of biologists interested in the evolution of sexual reproduction and its consequences for species survival.

**Decision letter after peer review:**

Thank you for submitting your article "Uniparental nuclear inheritance during bisexual mating in fungi" for consideration by *eLife*. Your article has been reviewed by 3 peer reviewers, one of whom is a member of our Board of Reviewing Editors, and the evaluation has been overseen by Patricia Wittkopp as the Senior Editor. The following individual involved in review of your submission has agreed to reveal their identity: Christina Hull (Reviewer #3).

Essential revisions:

1) It is not clear whether the observed process can be properly classified as hybridogenesis or not. While in some parts of the text the authors are very careful to refer to a process "similar" or "akin" to hybridogenesis, in others they plainly call it hybridogenesis. Yet, it appears that there are several reasons why this should not be referred to as hybridogenesis, among others: (i) no hybridization and (ii) some of the hallmarks that occur in hybridogenesis (e.g. endoreplication) are not shown or further discussed. Presenting this as a possible new form of sexual reproduction would be reasonable, but the authors would need to definitively show that (1) this is not unisexual reproduction as seen in C. deneoformans (easy to do – grow the strains in monoculture and evaluate for spore production) and (2) provide compelling data showing that mt inheritance is uniparental.

2) The reviewers suggest a series of points that would help improving the manuscript, making it accessible to a larger audience. In particular several of the clarifications revolve around one core aspect: is this hybridogenesis, or is it a new mechanism of sexual development for C. neoformans that more closely resembles hybridogenesis than other known forms of sexual development? To explicitly compare and contrast the features of both would be helpful.

3) It is important to clarify what is supported by direct evidence and what is not (i.e endoreplication?

4) Only a minimal set of experiments are suggested, particularly related to rule out unisexual reproduction, and to clearly support the uniparental inheritance of mt, while also providing clearer information on datasets (i.e germination rate H99alpha x IUM96a).

*Reviewer #1 (Recommendations for the authors):*

1) The authors state that "this process of uniparental sexual reproduction of C. neoformans exhibits striking parallels with hybridogenesis in animals". Yet, they don't explicitly comment what are the features/issues that would preclude them from claiming that the process is fully comparable to hybridogenesis.

Moreover, later on (page 10, line 19) they appear to indicate that the characterized phenomenon is, actually, hybridogenesis: "The discovery of hybridogenesis further diversifies known reproductive modes in fungi, suggesting the presence of sexual parasitism in this kingdom".

Thus, it is unclear throughout the text whether all the requirements to define this process as bona fide hybridogenesis are met or not.

2) An important aspect of hybridogenesis that is not explicitly discussed in the manuscript either, relates to the fact that hybridogenesis is normally defined as "an unusual form of reproduction that is found in hybrids between different species" (i.e PMID: 30721675). Yet, in the present manuscript the authors obtain their data in crosses between isolates that are part of the same species (albeit with enough genetic differences to clearly differentiate their nuclei, but below the threshold of species differences). This is not acknowledged or discussed in the text.

Moreover, the authors have recently published fine work in the context of hybrids between C. neoformans and C. deneoformans, and how species boundary (and conventional sexual reproduction) are stablished (PMID: 33465111). Indeed, it is quite relevant that such hybrids can be regularly identified from both environmental and clinical samples, in some cases representing a large percentage of the isolates (~20% of the isolates identified in Europe, with even higher prevalence in some EU countries -i.e. Portugal 31%- PMID: 32511709). Detecting hybridogenesis in such context would be a major thing, and would help putting the work into perspective. Indeed, it appears that hybridogenesis tends to occur between a hybrid and another species (related to the hybrid) which genome is then discarded. Thus, the reports of hybridogenesis in several animal species suggest that its origin may imply diverse hybridization events involving intergeneric paternal ancestors. None of these complexities and peculiarities are commented or discussed. Albeit hybridogenesis is not as "popular" as other unusual reproduction modes (i.e. Parthenogenesis), its existence in fungi would be a relevant finding, which would be better appreciated by a general audience if the basics of hybridogenesis are clearly stated, and a checklist of its basics is completed, based on the findings in C. neoformans.

3) It would be valuable that the authors would extend their discussions to other members of the Cryptococcus neoformans species complex. Indeed, the current evidence indicates that C. neoformans exhibits bisexual reproduction, whereas in C. deneoformans displays both bisexual and unisexual mating. Therefore, it would be interesting to see whether hybridogenesis appears as a peculiarity of C. neoformans, or as general property of the species complex (and particularly in the context of hybrids, see previous point). Likewise, the same question could be extended to the six members of the Cryptococcus gattii species complex. Nevertheless, as conducting those experiments would involve massive amount of work, the authors could expand the discussion of the topic (somehow covered already in page 11). In this context the authors could explicitly discuss whether they expect hybridogenesis to be a phenomenon that was acquired in C. neoformans, or a property that was lost in other members of the Cryptococcus species complex.

4) Page 6, Line 33: "Nuclear segregation can be followed by endoreplication occurring in these hyphal branches or in the basidia to produce a diploid nucleus that then ultimately undergoes meiosis and produces uniparental progeny"

It is not clear if with this phrase the authors are stating a possibility or a fact. What is the evidence that endoreplication is occurring?

Related to this, it is interesting that endoreplication has been observed as a one of the mechanisms of unisexual mating in C. deneoformans.

5) Page 8, Line 33: "Congruent with the mating-type analysis, the progeny exclusively inherited nuclear genetic material from only one of the two parents".

Please specify which one.

6) Page 10, Line 7: "These results combined with spore dissection data show that Dmc1 is critical for uniparental sporulation". Authors could enrich the phrase by explicitly comparing the requirement of meiotic recombinase-based mechanisms in animal hybridogenesis.

Page 11, Line 23: "As a result, hybridogenesis would result in the generation of predominantly progeny in nature reducing the MATa population and thus favoring the expansion of the mating-type population" While this is an interesting idea, it is also hard to ignore the fact that the hybridogenesis process described by the authors is rare (approx. 1 %) compared to classic sexual reproduction. Therefore, it would contribute to a minor fraction of the reproductive events occurring in nature, unless some environmental conditions facilitate the former mode over the latter.

*Reviewer #3 (Recommendations for the authors):*

Thank you to the authors for the opportunity to review their excellent and interesting work! I have several recommendations for the authors to consider.

1) This is an exciting discovery that could be of broad interest, but it is difficult to keep up with what the authors are trying to communicate in this manuscript. Specifically, the terminology to refer to the various forms of sexual development and spore production (i.e. uniparental sporulation, uniparental nuclear inheritance, bisexual reproduction, unisexual sporulation, etc.) are not well defined and used inconsistently. The challenge is confounded by the (necessary) inclusion in the mix of "uniparental mitochondrial inheritance." The authors could improve the manuscript substantially by carefully defining and consistently using intentional labels for the numerous, relevant processes (including "mating," which is best used when referring to only the fusion process and not all of sexual development). It will help readers understand their beautiful data.

2) A second point of possible confusion is the use of "Germination Rate" to refer to the proportion of spores that can germinate into viable progeny. In the case of spores, it is probably more accurate to refer to this as "Germination Frequency" (high vs. low). Germination Rate is perhaps better used to indicate the efficiency with which any given spore differentiates into a yeast over time (slow vs. fast).

3) The manuscript would also benefit from careful editing so that all figures are referenced in the text, figure legends and the text provide the same information, and the prose all makes sense. Some specific problems are indicated below.

4) With respect to the mitochondrial inheritance data, Cryptococcus researchers likely know how one determines the parental source of the mitochondria in F1 progeny, but others may not. It is also challenging because all of the progeny (with only one exception) harbor "a"-derived mitochondria. It is a somewhat annoying but necessary question: How do you know your primers can discern between "a" and α-derived mitochondria? What is the basis for this discrimination? What if the primers were mislabeled – would you know? More information and context in the manuscript would help resolve any doubts.

5) After Figure 2, there is a leap and a disconnect. The authors show clear evidence of one nuclear loss event at a hyphal branch, supporting their model. Then they indicate that endoreplication and meiosis can occur but do not provide any data for this. Then the manuscript addresses the presence or absence of meiotic recombination and the role of DMC1 in uniparental nuclear inheritance. It is not clear how this transition is made – particularly if one is outside the Cryptococcus field and does not know about the events that occur during unisexual reproduction in C. deneoformans. The absence of meiotic recombination in uniparental nuclear inheritance would make sense, but then why investigate a meiotic recombinase subsequent to that? Providing context for the DMC1 experiments would be extremely useful, as would including an interpretation of the findings in the Discussion.

6) Food for thought: What if loss of one parent nucleus is just a mistake? What if, at some frequency, in all forms of Cryptococcus reproduction clamp cells mess up and a nucleus gets lost (maybe more at branchpoints) but it "just happens" in ~1% of all filaments. Is that really hybridosis or akin to hybridosis? It seems a little risky to liken a very rare event that occurs during sexual development to a form of animal development that is the primary mechanism of reproduction within certain animal species. Is it then reasonable to posit that understanding the seemingly very rare uniparental nuclear inheritance process in a fungus will inform sexual evolution in larger eukaryotes? On the other hand, if the C. neoformans response to losing one of the parental nuclei is endoreplication (as the authors suggest but do not show), that seems to be more of a potential parallel.

How likely is this very low frequency event to contribute to fitness if it is occurring coincident with bisexual reproduction, as suggested by the authors? Perhaps it contributes to fitness, but perhaps not. Can such a rare event be reasonably compared to sexual parasitism?

Page 3, line 1: "Most organisms in nature undergo sexual reproduction between two partners of the opposite sex to produce progeny." Is it most? There are a lot of microbes in the world that don't undergo sexual reproduction as described here.

Page 5, line 17: At that point, how do you know it's sexual?

Page 9, lines 26-30: "Dmc1 mutant unilateral crosses sporulated at a frequency of

0.4% suggesting that only one of the parental strains was producing spores (Figure 4B). When a few sporulating basidia from multiple mating spots were dissected, two different populations of basidia emerged, one with no spore germination, and the other with a high spore germination rate and uniparental DNA inheritance (Table 2)." Something here seems amiss – why are there two populations of spores? Perhaps re-phrasing could help clarify.

Page 9, line 24: Is the presence of both MAT alleles indicative of aneuploidy? Couldn't it also be diploidy? Also, need to change "much fewer" to "many fewer."

Is bisexual reproduction normal in nature?

There is no specific reference to Figure S2B in the manuscript.

Figure 2 Legend refers to unisexual reproduction – should be uniparental? This happens several times.

Labels for which consolidation and/or definitions would be helpful:

From Title and Abstract:

Uniparental nuclear inheritance

Bisexual mating

Bisexual reproduction

Uniparental sporulation

Uniparental reproduction

From Introduction:

Sexual reproduction

Unisexual reproduction

Heterothallic sexual reproduction

Mitochondrial Uniparental Inheritance

Uniparental inheritance of nuclei

Partner-stimulated uniparental sexual reproduction

From Results:

Unusual sexual reproduction

Uniparental mitochondrial genome inheritance

Unisexual reproduction

Uniparental inheritance

Uniparental sporulation

Uniparental reproduction

Biparental sporulation

Uniparental meiosis and sporulation

Uniparental MAT inheritance

this unusual mode of unisexual reproduction occurs in nature in parallel with normal bisexual reproduction

Bilateral crosses

Unilateral crosses

[Editors' note: further revisions were suggested prior to acceptance, as described below.]

Thank you for resubmitting your work entitled "Uniparental nuclear inheritance following bisexual mating in fungi" for further consideration by *eLife*. Your revised article has been evaluated by 3 peer reviewers, one of whom is a member of our Board of Reviewing Editors, and the evaluation has been overseen by Patricia Wittkopp (Senior Editor).

The manuscript has been improved but there are some remaining issues that need to be addressed, as outlined below:

I am confident that incorporating the different suggestions can be easily done and should not take much time.

*Reviewer #1 (Recommendations for the authors):*

The revised manuscript has included key recommendations previously pointed by reviewers. In particular, the authors have been more conservative in classifying the observed phenomenon as pseudosexual reproduction, which shares several features of hybridogenesis, albeit it may be too soon to classify it unequivocally us such. They also have provided additional data solidifying their conclusions about mitochondria inheritance. These findings, described for the first time in the fungal kingdom, may provide important insights for understanding the evolution of sex. Indeed, this pseudosexual reproduction (particularly related to genome exclusion followed by endoreplication and meiosis) could correspond to an ancestral form of reproduction that preceded the evolution of sexual reproduction. While the fine mechanisms allowing pseudosexual reproduction still remain undefined (i.e. endoreplication is suspected to occur, although no experiments actually confirm or deny its occurrence), the authors acknowledge the current limitations of the study.

*Reviewer #3 (Recommendations for the authors):*

In the resubmitted manuscript "Uniparental nuclear inheritance following bisexual mating in fungi," the authors offer a thorough and thoughtful response to reviewer criticisms and extensive revisions to the paper. Through their efforts, the manuscript is substantially improved – offering a more accurate presentation of the data, more clarity in the data presentation, and consideration of points that will make the findings more accessible to a broad audience of researchers.

The authors were particularly responsive to the criticism that the term hybridogenesis might not apply to their findings. They modified the manuscript accordingly by providing pivotal evidence to rule out unisexual reproduction, engaging in a richer consideration of alternative hypotheses, and clarifying/unifying their use of language to refer to sexual processes. As a result, the data are more clear, and the arguments are more compelling.

Points for consideration:

Figure 4: It is still unclear how many times the authors determined that nuclear loss occurred at a branchpoint during hyphal growth in pseudosexual reproduction. I recognize that capturing these events is difficult, so a reference to the images shown as being representative of what is seen in Figure 4 and a general accounting of the number of times this pattern has been observed would be useful (in text and/or legend).

Figure 6: In terms of understanding the mechanism(s) by which pseudosexual reproduction could occur and influence fitness, Figure 6A could be eliminated (and referred to in supplemental). Figure 6B could be retained as simply Figure 6 and be referred to in the Discussion.

Throughout text:

Clarity of the manuscript could be improved even further by using the term "mating" to refer only to the fusion event between parents (the first step of sexual development) and not to the entire process of development.

*Reviewer #4 (Recommendations for the authors):*

The authors have responded satisfactorily to the comments from the previous review.

I have some additional suggestions for the authors:

1. The title might be improved to indicate that the uniparental nuclear genome inheritance enables nuclear-mitochondrial genome swapping, which the authors both demonstrate and is a key piece of the mechanistic argument.

2. The authors may want to examine more cases of the observed phenomenon before concluding that a particular mating type is the favored result of the pseudo sexual process. In other words, it may not be mating type but rather nuclear genotype more generally that determines which nucleus is lost during dikaryotic development.

3. The authors may want to compare their observations with cytoduction in *S. cerevisiae*.

---

## [Author Response]

Essential Revisions (for the authors):1) It is not clear whether the observed process can be properly classified as hybridogenesis or not. While in some parts of the text the authors are very careful to refer to a process "similar" or "akin" to hybridogenesis, in others they plainly call it hybridogenesis. Yet, it appears that there are several reasons why this should not be referred to as hybridogenesis, among others: i) no hybridization and ii) some of the hallmarks that occur in hybridogenesis (e.g. endoreplication) are not shown or further discussed. Presenting this as a possible new form of sexual reproduction would be reasonable, but the authors would need to definitively show that 1) this is not unisexual reproduction as seen in C. deneoformans (easy to do – grow the strains in monoculture and evaluate for spore production) and 2) provide compelling data showing that mt inheritance is uniparental.

We thank the editor and reviewers for their insightful and critical comments. While some of the crosses analyzed in our study could be considered as hybridization given the significant structural chromosomal rearrangements between the two isolates (e.g. VYD135 vs. KN99a), we agree that the novel phenomenon of the production of clonal progeny with a uniparental nuclear inheritance could also occur between strains that are genetically closely related. Thus, we have decided to refrain from calling it hybridogenesis. Instead, we now define it as a novel form of reproduction and term it “pseudosexual reproduction”, reflecting the finding that although the observed process starts with bisexual mating, the two parental nuclei do not recombine and the progeny are clonal with the nuclear genome that is identical to one of the two parents.

Following the reviewers’ suggestions, we included compelling additional data definitively demonstrating mitochondrial uniparental inheritance during pseudosexual reproduction by both PCR as well as whole-genome sequencing (Figures 1, 4, Figure 3—figure supplement 1, and Figure 3—figure supplement 4).

We also included images for monocultures of the strains used in our crosses under sex-inducing conditions, confirming that none of them is capable of selfing and thus excluding the possibility that the observed process is unisexual reproduction (Figures 1 and Figure 3—figure supplement 1).

Moreover, our microscopic observation of fluorescently labelled strains of opposite mating type undergoing sexual reproduction provides additional evidence that hyphae containing both parental nuclei experience nuclear mis-segration at branch points, resulting in more distal hyphal compartments and basidia containing only one or the other parental nucleus.

Taken together, these findings exclude unisexual reproduction as an explanation for our observations of an unusual sexual cycle involving two parents but giving rise to clonal progeny with nuclear genomes related to only one parental nuclear genotype.

We revised our manuscript accordingly and hope that the editor and reviewers find these revisions satisfactory.

2) The reviewers suggest a series of points that would help improving the manuscript, making it accessible to a larger audience. In particular several of the clarifications revolve around one core aspect: is this hybridogenesis, or is it a new mechanism of sexual development for C. neoformans that more closely resembles hybridogenesis than other known forms of sexual development? To explicitly compare and contrast the features of both would be helpful.

This is a great suggestion! We revised our manuscript thoroughly and included additional discussion. As mentioned in our previous response, we now consider the phenomenon that we discovered represents a novel form of sexual reproduction that we have termed pseudosexual reproduction, which does share similarities with hybridogenesis. We have also revised the discussion to provide a detailed comparison of the two processes, pseudosexual reproduction and hybridogenesis.

3) It is important to clarify what is supported by direct evidence and what is not (i.e endoreplication?

We revised the manuscript accordingly to explicitly state the results and hypothesis. While our experiments are consistent with and suggestive of endoreplication, we do not have direct evidence to state this unequivocally. We revised the text to clarify that the presence of endoreplication currently is only a hypothesis. While we do think endoreplication is likely occurring during pseudosexual reproduction, given that our molecular studies provide evidence of meiosis resulting in the production of four spore chains, other mechanisms are possible and assessing these requires further investigation, which is beyond the scope of the current study.

4) Only a minimal set of experiments are suggested, particularly related to rule out unisexual reproduction, and to clearly support the uniparental inheritance of mt, while also providing clearer information on datasets (i.e germination rate H99alpha x IUM96a).

We thank the editor and reviewers for the opportunity to revise and resubmit the manuscript. As mentioned above, we have provided additional data to rule out unisexual reproduction and definitively supporting uniparental mitochondrial inheritance (Figures 1, 4, Figure 3—figure supplement 1, and Figure 3—figure supplement 4). We also revised the discussion to include more details as suggested by the reviewers, for example with respect to germination rates.

Reviewer #1 (Recommendations for the authors):1) The authors state that "this process of uniparental sexual reproduction of C. neoformans exhibits striking parallels with hybridogenesis in animals". Yet, they don't explicitly comment what are the features/issues that would preclude them from claiming that the process is fully comparable to hybridogenesis.Moreover, later on (page 10, line 19) they appear to indicate that the characterized phenomenon is, actually, hybridogenesis: "The discovery of hybridogenesis further diversifies known reproductive modes in fungi, suggesting the presence of sexual parasitism in this kingdom".Thus, it is unclear throughout the text whether all the requirements to define this process as bona fide hybridogenesis are met or not.

We appreciate the reviewer’s comment. As mentioned in responses to previous comments from the other reviewers and editor, we now consider the mode of reproduction discovered in our study as a novel form of sexual reproduction and term it as pseudosexual reproduction, which resembles hybridogenesis. We have revised the manuscript accordingly and included a detailed comparison between pseudosexual reproduction and hybridogenesis. We hope the reviewer finds our revision now satisfactory.

2) An important aspect of hybridogenesis that is not explicitly discussed in the manuscript either, relates to the fact that hybridogenesis is normally defined as "an unusual form of reproduction that is found in hybrids between different species" (i.e PMID: 30721675). Yet, in the present manuscript the authors obtain their data in crosses between isolates that are part of the same species (albeit with enough genetic differences to clearly differentiate their nuclei, but below the threshold of species differences). This is not acknowledged or discussed in the text.Moreover, the authors have recently published fine work in the context of hybrids between C. neoformans and C. deneoformans, and how species boundary (and conventional sexual reproduction) are stablished (PMID: 33465111). Indeed, it is quite relevant that such hybrids can be regularly identified from both environmental and clinical samples, in some cases representing a large percentage of the isolates (~20% of the isolates identified in Europe, with even higher prevalence in some EU countries -i.e. Portugal 31%- PMID: 32511709). Detecting hybridogenesis in such context would be a major thing, and would help putting the work into perspective. Indeed, it appears that hybridogenesis tends to occur between a hybrid and another species (related to the hybrid) which genome is then discarded. Thus, the reports of hybridogenesis in several animal species suggest that its origin may imply diverse hybridization events involving intergeneric paternal ancestors. None of these complexities and peculiarities are commented or discussed. Albeit hybridogenesis is not as "popular" as other unusual reproduction modes (i.e. Parthenogenesis), its existence in fungi would be a relevant finding, which would be better appreciated by a general audience if the basics of hybridogenesis are clearly stated, and a checklist of its basics is completed, based on the findings in C. neoformans.

We thank the reviewer for a great suggestion! We have revised our manuscript accordingly. Specifically, we made it clear that the described phenomenon is not yet reported in hybrids of Cryptococcus neoformans and Cryptococcus deneoformans and represents a novel form of reproduction that resembles hybridogenesis. We also provided a detailed comparison between pseudosexual reproduction and hybridogenesis as suggested by the reviewer. We are currently expanding this work to study whether pseudosexual reproduction occurs during sexual reproduction of other Cryptococcus species, as well as during hybridization between different species.

3) It would be valuable that the authors would extend their discussions to other members of the Cryptococcus neoformans species complex. Indeed, the current evidence indicates that C. neoformans exhibits bisexual reproduction, whereas in C. deneoformans displays both bisexual and unisexual mating. Therefore, it would be interesting to see whether hybridogenesis appears as a peculiarity of C. neoformans, or as general property of the species complex (and particularly in the context of hybrids, see previous point). Likewise, the same question could be extended to the six members of the Cryptococcus gattii species complex. Nevertheless, as conducting those experiments would involve massive amount of work, the authors could expand the discussion of the topic (somehow covered already in page 11). In this context the authors could explicitly discuss whether they expect hybridogenesis to be a phenomenon that was acquired in C. neoformans, or a property that was lost in other members of the Cryptococcus species complex.

We are currently studying the occurrence of this new mode of reproduction in other species of Cryptococcus. We aim to conduct those experiments and answer some of the questions raised here in the next study. While we did not include any data from these experiments in this study, we added relevant discussion in the revised manuscript. We also included a model figure (Figure 6A) to differentiate between unisexual and pseudosexual reproduction along with a more thorough comparison in the revised manuscript.

4) Page 6, Line 33: "Nuclear segregation can be followed by endoreplication occurring in these hyphal branches or in the basidia to produce a diploid nucleus that then ultimately undergoes meiosis and produces uniparental progeny"It is not clear if with this phrase the authors are stating a possibility or a fact. What is the evidence that endoreplication is occurring?Related to this, it is interesting that endoreplication has been observed as a one of the mechanisms of unisexual mating in C. deneoformans.

While we suspect the occurrence of endoreplication after nuclear segregation, we do not have concrete evidence to support this. Thus, we revised the manuscript suggesting endoreplication as a possible mechanism, which requires further study. As the reviewer mentioned, the occurrence of endoreplication in C. deneoformans during unisexual reproduction raises the possibility that C. neoformans might similarly undergo endoreplication during pseudosexual reproduction. We included this possibility in the revised manuscript.

5) Page 8, Line 33: "Congruent with the mating-type analysis, the progeny exclusively inherited nuclear genetic material from only one of the two parents".Please specify which one.

The referred statement is about VYD135α x IUM96a cross where we observed inheritance of MATα nuclei in progeny from all of the basidia, with exception of one in which all of the progeny inherited the nuclear genome from the MATa parent. Given that, none of the parental strains is capable of selfing; our data suggest pseudosexual reproduction can involve either of the two parental nuclei to produce uniparental progeny. We revised the text to make this point clearer.

6) Page 10, Line 7: "These results combined with spore dissection data show that Dmc1 is critical for uniparental sporulation". Authors could enrich the phrase by explicitly comparing the requirement of meiotic recombinase-based mechanisms in animal hybridogenesis.

This is a great suggestion! We conducted a thorough search in the literature on studies of hybridogenesis. While we could not find any study suggesting the requirement for a meiotic recombinase, we did find previous studies showing synaptonemal complex formation as evidence of meiosis during hybridogenesis. We added discussion of these studies in the revised manuscript.

Page 11, Line 23: "As a result, hybridogenesis would result in the generation of predominantly progeny in nature reducing the MATa population and thus favoring the expansion of the mating-type population" While this is an interesting idea, it is also hard to ignore the fact that the hybridogenesis process described by the authors is rare (approx. 1 %) compared to classic sexual reproduction. Therefore, it would contribute to a minor fraction of the reproductive events occurring in nature, unless some environmental conditions facilitate the former mode over the latter.

We agree with the reviewer that 1% of sporulation events under normal conditions might not be considered to be of significance. However, we discuss a few hypothetical scenarios where even a 1% successful sporulation rate could be of great advantage as compared to sexual reproduction events (Figure 6B). Unfortunately, Cryptococcus mating occurs under highly specific conditions restricting testing of variable environmental conditions. At the same time, we cannot rule out that there might be some yet-to-be-identified natural conditions that facilitate pseudosexual reproduction and might even promote it.

Reviewer #3 (Recommendations for the authors):Thank you to the authors for the opportunity to review their excellent and interesting work! I have several recommendations for the authors to consider.1) This is an exciting discovery that could be of broad interest, but it is difficult to keep up with what the authors are trying to communicate in this manuscript. Specifically, the terminology to refer to the various forms of sexual development and spore production (i.e. uniparental sporulation, uniparental nuclear inheritance, bisexual reproduction, unisexual sporulation, etc.) are not well defined and used inconsistently. The challenge is confounded by the (necessary) inclusion in the mix of "uniparental mitochondrial inheritance." The authors could improve the manuscript substantially by carefully defining and consistently using intentional labels for the numerous, relevant processes (including "mating," which is best used when referring to only the fusion process and not all of sexual development). It will help readers understand their beautiful data.

We thank the reviewer for their very useful comment. As recommended, we have removed multiple terms from the manuscript and defined the ones that we have retained. We also revised our manuscript thoroughly to make sure that we use the same term for a defined process consistently, thus avoiding multiple terms. We hope the reviewer will find these revisions satisfactory.

2) A second point of possible confusion is the use of "Germination Rate" to refer to the proportion of spores that can germinate into viable progeny. In the case of spores, it is probably more accurate to refer to this as "Germination Frequency" (high vs. low). Germination Rate is perhaps better used to indicate the efficiency with which any given spore differentiates into a yeast over time (slow vs. fast).

We modified the text as suggested by the reviewer.

3) The manuscript would also benefit from careful editing so that all figures are referenced in the text, figure legends and the text provide the same information, and the prose all makes sense. Some specific problems are indicated below.

We revised the manuscript to include all necessary details and referred to all of the data presented.

4) With respect to the mitochondrial inheritance data, Cryptococcus researchers likely know how one determines the parental source of the mitochondria in F1 progeny, but others may not. It is also challenging because all of the progeny (with only one exception) harbor "a"-derived mitochondria. It is a somewhat annoying but necessary question: How do you know your primers can discern between "a" and α-derived mitochondria? What is the basis for this discrimination? What if the primers were mislabeled – would you know? More information and context in the manuscript would help resolve any doubts.

We included additional and clearer details on genotyping of mitochondrial DNA in the revised manuscript. We have provided the schemes of alleles that were employed to discern MATα mitochondria versus MATa mitochondria (Figures 1 and Figure 3—figure supplement 1). The maps also include the locations of primers to indicate the length of amplicons. Furthermore, we included images of the PCR assay to show the controls and results from our genotyping data.

5) After Figure 2, there is a leap and a disconnect. The authors show clear evidence of one nuclear loss event at a hyphal branch, supporting their model. Then they indicate that endoreplication and meiosis can occur but do not provide any data for this. Then the manuscript addresses the presence or absence of meiotic recombination and the role of DMC1 in uniparental nuclear inheritance. It is not clear how this transition is made – particularly if one is outside the Cryptococcus field and does not know about the events that occur during unisexual reproduction in C. deneoformans. The absence of meiotic recombination in uniparental nuclear inheritance would make sense, but then why investigate a meiotic recombinase subsequent to that? Providing context for the DMC1 experiments would be extremely useful, as would including an interpretation of the findings in the Discussion.

We revised the manuscript to alter the flow of information in the Results section. We now first show results suggesting the occurrence of uniparental nuclear inheritance in lab strains. Then we show the occurrence of the same phenomenon in the natural isolates along with the evidence that two parental genomes do not mix with each other in the generation of the uniparental progeny. Then, we delve into the mechanism that shows that the two nuclei separate during hyphal branching. In the end, we show that the meiotic recombinase DMC1 is required for the generation of uniparental progeny, providing evidence that meiosis is still a key part of the uniparental nuclear inheritance. This also suggests that the remaining nucleus undergoes endoreplication or another process of genome duplication. We included additional text to provide a better context for the Dmc1 experiments and revised the discussion accordingly.

6) Food for thought: What if loss of one parent nucleus is just a mistake? What if, at some frequency, in all forms of Cryptococcus reproduction clamp cells mess up and a nucleus gets lost (maybe more at branchpoints) but it "just happens" in ~1% of all filaments. Is that really hybridosis or akin to hybridosis? It seems a little risky to liken a very rare event that occurs during sexual development to a form of animal development that is the primary mechanism of reproduction within certain animal species. Is it then reasonable to posit that understanding the seemingly very rare uniparental nuclear inheritance process in a fungus will inform sexual evolution in larger eukaryotes? On the other hand, if the C. neoformans response to losing one of the parental nuclei is endoreplication (as the authors suggest but do not show), that seems to be more of a potential parallel.

We appreciate the reviewer’s very insightful comment. Our initial hypothesis included the possibility that this process is occurring due to some errors. However, a 1% error rate is probably too high to be explained by nuclear segregation errors or clamp cells. Therefore, we think the phenomenon observed cannot be entirely explained by these errors. If multiple errors are contributing to the phenomenon, it is important to understand the process and the causal factors. While we accept the reviewer’s point of view, we think that this is a novel reproductive process and could have significant implications for Cryptococcus biology.

We agree that this may not be hybridogenesis as such and we have revised the manuscript to suggest that this is a novel form of reproduction in Cryptococcus neoformans. However, the two processes share many similarities and we cannot rule out the possibility that studying one will help in understanding the other. Future studies will provide more clarity on the connection between these two phenomena.

We agree with the reviewer that endoreplication might be a response of C. neoformans to loss of one of the nuclei. If so, we are not sure we could call this process a result of errors alone. Also, what might seem like an error to us could be a survival strategy for C. neoformans. In other words, pseudosexual reproduction may be a well-programmed mechanism that enables C. neoformans to survive under unfavorable conditions.

We included some of these hypothetical scenarios where uniparental nuclear inheritance might be more beneficial than sexual reproduction (Figure 6B). Thus, understanding this process might provide important insights into the evolution of this human fungal pathogen.

How likely is this very low frequency event to contribute to fitness if it is occurring coincident with bisexual reproduction, as suggested by the authors? Perhaps it contributes to fitness, but perhaps not. Can such a rare event be reasonably compared to sexual parasitism?

Albeit at low frequency, pseudosexual reproduction can be beneficial in certain conditions when the two parents can initiate mating, but are not able to complete classic sexual reproduction due to genetic incompatibilities such as sequence divergence, chromosomal structural variation, or a combination of the two. We have revised the manuscript to include a discussion of multiple possible scenarios (Figure 6B and discussion). While it cannot be compared explicitly to sexual parasitism, their outcomes are similar.

Page 3, line 1: "Most organisms in nature undergo sexual reproduction between two partners of the opposite sex to produce progeny." Is it most? There are a lot of microbes in the world that don't undergo sexual reproduction as described here.

We modified the sentence as suggested by the reviewer.

Page 5, line 17: At that point, how do you know it's sexual?

We modified the text in the revised manuscript to make this point clearer.

Page 9, lines 26-30: "Dmc1 mutant unilateral crosses sporulated at a frequency of0.4% suggesting that only one of the parental strains was producing spores (Figure 4B). When a few sporulating basidia from multiple mating spots were dissected, two different populations of basidia emerged, one with no spore germination, and the other with a high spore germination rate and uniparental DNA inheritance (Table 2)." Something here seems amiss – why are there two populations of spores? Perhaps re-phrasing could help clarify.

We revised the text to make this clearer. The two populations most likely arise because some basidia can bypass the requirement for Dmc1 while most others cannot.

Page 9, line 24: Is the presence of both MAT alleles indicative of aneuploidy? Couldn't it also be diploidy? Also, need to change "much fewer" to "many fewer."

We thank the reviewer for bringing this to our attention. The presence of both MAT alleles could be due to diploidy as well. We modified the text as suggested by the reviewer.

Is bisexual reproduction normal in nature?

We think the reviewer may be referring to the presence of bisexual reproduction in Cryptococcus neoformans with this point. While there is no direct evidence for bisexual reproduction in C. neoformans in nature, several lines of evidence suggest that this is likely occurring, at least in certain areas. First, many natural isolates are capable of undergoing sexual reproduction in the lab. Second, while there is an overall bias toward the α mating type among natural isolates, in certain geographic regions the distribution of the two mating types in the natural population is more balanced, suggesting the opportunity for sexual reproduction is present in nature. Third, population genomics studies of the strains isolated from regions with balanced mating types show clear signatures of recombination at the population level. Fourth, naturally occuring inter-species AD hybrids are diploid and most often harbor MAT alleles of opposite mating type. Finally, sexual reproduction has been shown to occur on plants, or on pigeon guano media, two common natural habitats of Cryptococcus.

There is no specific reference to Figure S2B in the manuscript.

We previously referred to this figure as “Figure S2” (now Figure 2—figure supplement 1) in the second Results section, which was meant to refer to both panels A and B. We modified the text to specifically refer to these as “Figure 3—figure supplement 1A and B” in the revised draft.

Figure 2 Legend refers to unisexual reproduction – should be uniparental? This happens several times.

We have extensively revised the manuscript to avoid any confusion arising due to different terminology. We have entirely refrained from using the term “unisexual” for the process described in this study.

Labels for which consolidation and/or definitions would be helpful:From Title and Abstract:Uniparental nuclear inheritanceBisexual matingBisexual reproductionUniparental sporulationUniparental reproductionFrom Introduction:Sexual reproductionUnisexual reproductionHeterothallic sexual reproductionMitochondrial Uniparental InheritanceUniparental inheritance of nucleiPartner-stimulated uniparental sexual reproductionFrom Results:Unusual sexual reproductionUniparental mitochondrial genome inheritanceUnisexual reproductionUniparental inheritanceUniparental sporulationUniparental reproductionBiparental sporulationUniparental meiosis and sporulationUniparental MAT inheritancethis unusual mode of unisexual reproduction occurs in nature in parallel with normal bisexual reproductionBilateral crossesUnilateral crosses

We thank the reviewer for bringing this to our attention. We now revised the manuscript, reduced the terms used throughout the manuscript, and defined the ones that are being used. We really appreciate the reviewer’s effort in helping us to simplify the manuscript and clarify the text.

[Editors' note: further revisions were suggested prior to acceptance, as described below.]

Reviewer #1 (Recommendations for the authors):The revised manuscript has included key recommendations previously pointed by reviewers. In particular, the authors have been more conservative in classifying the observed phenomenon as pseudosexual reproduction, which shares several features of hybridogenesis, albeit it may be too soon to classify it unequivocally us such. They also have provided additional data solidifying their conclusions about mitochondria inheritance. These findings, described for the first time in the fungal kingdom, may provide important insights for understanding the evolution of sex. Indeed, this pseudosexual reproduction (particularly related to genome exclusion followed by endoreplication and meiosis) could correspond to an ancestral form of reproduction that preceded the evolution of sexual reproduction. While the fine mechanisms allowing pseudosexual reproduction still remain undefined (i.e. endoreplication is suspected to occur, although no experiments actually confirm or deny its occurrence), the authors acknowledge the current limitations of the study.

We are once again thankful for the insightful and critical review that helped us significantly improve the clarity and presentation of the manuscript.

Reviewer #3 (Recommendations for the authors):In the resubmitted manuscript "Uniparental nuclear inheritance following bisexual mating in fungi," the authors offer a thorough and thoughtful response to reviewer criticisms and extensive revisions to the paper. Through their efforts, the manuscript is substantially improved – offering a more accurate presentation of the data, more clarity in the data presentation, and consideration of points that will make the findings more accessible to a broad audience of researchers.The authors were particularly responsive to the criticism that the term hybridogenesis might not apply to their findings. They modified the manuscript accordingly by providing pivotal evidence to rule out unisexual reproduction, engaging in a richer consideration of alternative hypotheses, and clarifying/unifying their use of language to refer to sexual processes. As a result, the data are more clear, and the arguments are more compelling.

We would like to extend our sincere thanks to the reviewers for their insights and thoughtful suggestions. Their comments helped us in significantly improving the manuscript and making it both clearer and, we think, more compelling.

Points for consideration:Figure 4: It is still unclear how many times the authors determined that nuclear loss occurred at a branchpoint during hyphal growth in pseudosexual reproduction. I recognize that capturing these events is difficult, so a reference to the images shown as being representative of what is seen in Figure 4 and a general accounting of the number of times this pattern has been observed would be useful (in text and/or legend).

We have now revised the text to include specific details and modified the paragraph accordingly. The paragraph now reads:

“…Unfortunately, a majority of the hyphae (>30 independent hyphae) we tracked were embedded into the agar, and most of these could not be tracked to the point of branching. For some others, we were able to image the hyphal branching point where two nuclei separate from each other but were then either broken or did not have mature basidia on them (Figure 4—figure supplement 1B). In total, we observed seven events of nuclear loss at hyphal branching in independent experiments and were able to track two of them to observe sporulation or basidia formation at the tip. We also observed long hyphae with only one parental nucleus in the VYD135α x Bt63a cross as well, suggesting the mechanism might be similar between strains.”

Figure 6: In terms of understanding the mechanism(s) by which pseudosexual reproduction could occur and influence fitness, Figure 6A could be eliminated (and referred to in supplemental). Figure 6B could be retained as simply Figure 6 and be referred to in the Discussion.

We modified the figure as suggested by the reviewer. The original Figure 6A is now included as Figure 6—figure supplement 1 and the original figure 6B is now main Figure 6.

Throughout text:Clarity of the manuscript could be improved even further by using the term "mating" to refer only to the fusion event between parents (the first step of sexual development) and not to the entire process of development.

This is a great suggestion. We modified the text to only use the term “mating” for cell-cell fusion. Everywhere else, the term has been modified according to the reviewer’s suggestions provided in the marked PDF file.

Reviewer #4 (Recommendations for the authors):The authors have responded satisfactorily to the comments from the previous review.I have some additional suggestions for the authors:1. The title might be improved to indicate that the uniparental nuclear genome inheritance enables nuclear-mitochondrial genome swapping, which the authors both demonstrate and is a key piece of the mechanistic argument.

We thank the reviewer for highlighting this point. While it is indeed the case that pseudosexual reproduction enables nuclear-mitochondrial genome exchange in the MATα progeny, it does not do so in the MATa progeny. Thus, we added a sentence in the abstract to highlight this point, but have kept the same title in the revised manuscript. The modified abstract now reads as follows:

“…Pseudosexual reproduction was also detected in natural isolate crosses where it resulted in mainly MATα progeny, a bias observed in Cryptococcus ecological distribution as well. The mitochondria in these progeny were inherited from the MATa parent, resulting in nuclear-mitochondrial genome exchange….”

2. The authors may want to examine more cases of the observed phenomenon before concluding that a particular mating type is the favored result of the pseudo sexual process. In other words, it may not be mating type but rather nuclear genotype more generally that determines which nucleus is lost during dikaryotic development.

We have modified the text to clarify that this is a possible hypothesis and not a conclusion. The text now reads:

“…We hypothesize that MATa isolates may be defective in this process due to either a variation in their genomes or some other as yet undefined sporulation factor. As a result, pseudosexual reproduction could lead to the generation of predominantly α progeny in nature, reducing the MATa population and thus favoring the expansion of the α mating-type population. However, it is still possible that the preferential inheritance of the nuclear genome from one of the two parents is decided by genetic elements located in regions other than MAT, and whether the uniparental nuclear inheritance is mating-type specific remains to be elucidated….”

3. The authors may want to compare their observations with cytoduction in S. cerevisiae.

We thank the reviewer for this excellent suggestion. We added a paragraph in the discussion about cytoduction that reads as follows:

“One of the key differences between pseudosexual reproduction and unisexual reproduction observed in the *Cryptococcus* species complex is the inheritance of mitochondrial DNA. While both unisexual and pseudosexual reproduction result in clonal progeny with respect to the nuclear genome, the mitochondria in pseudosexual reproduction are almost exclusively inherited from the MATa parent (Figure 6—figure supplement 1). This results in the exchange of mitochondrial DNA in the progeny that inherit the MATα nuclear genome, resembling the nuclear-mitochondrial exchange observed during cytoduction in *Saccharomyces cerevisiae*. During cytoduction, mutants defective in nuclear fusion produce haploid progeny with nuclear genome from one parent, but a mixture of both parents cytoplasm resulting in the inheritance of one parental mitochondrial genome with the other parent’s nuclear genome (Conde and Fink, 1976; Lancashire and Mattoon, 1979; Zakharov and Yarovoy, 1977). This process was used to study mitochondrial genetics with respect to the transfer of drug-resistance genes and other mitochondrial mutations. Similar to cytoduction, pseudosexual reproduction could be employed to study mitochondrial genetics, such as functional analysis of mitochondrial encoded drug resistance, and cytoplasmic inheritance of factors such as prions in *C. neoformans*.”